# A scaffold lncRNA shapes the mitosis to meiosis switch

Vedrana Andric [1,6], Alicia Nevers[1,4,6], Ditipriya Hazra[2,5,6], Sylvie Auxilien [1], Alexandra Menant[1],
Marc Graille [2], Benoit Palancade [3] & Mathieu Rougemaille [1✉]

Long non-coding RNAs (lncRNAs) contribute to the regulation of gene expression in response to intra- or extracellular signals but the underlying molecular mechanisms remain largely unexplored. Here, we identify an uncharacterized lncRNA as a central player in shaping the meiotic gene expression program in fission yeast. We report that this regulatory RNA, termed *mamRNA*, scaffolds the antagonistic RNA-binding proteins Mmi1 and Mei2 to ensure their reciprocal inhibition and fine tune meiotic mRNA degradation during mitotic growth. Mechanistically, *mamRNA* allows Mmi1 to target Mei2 for ubiquitin-mediated downregulation, and conversely enables accumulating Mei2 to impede Mmi1 activity, thereby reinforcing the mitosis to meiosis switch. These regulations also occur within a unique Mmi1-containing nuclear body, positioning *mamRNA* as a spatially-confined sensor of Mei2 levels. Our results thus provide a mechanistic basis for the mutual control of gametogenesis effectors and further expand our vision of the regulatory potential of lncRNAs.

[1] Université Paris-Saclay, CEA, CNRS, Institute for Integrative Biology of the Cell (I2BC), 91198 Gif-sur-Yvette, France. [2] Laboratoire de Biologie Structurale de la Cellule (BIOC), CNRS, Ecole polytechnique, Institut Polytechnique de Paris, 91128 Palaiseau, France. [3] Université de Paris, CNRS, Institut Jacques Monod, F-75006 Paris, France. [4] Present address: University Paris-Saclay, INRAE, AgroParisTech, Micalis Institute, 78350 Jouy-en-Josas, France. [5] Present address: Department of Biochemistry, Oxford University, Oxford OX1 3QU, UK. [6] These authors contributed equally: Vedrana Andric, Alicia Nevers, Ditipriya Hazra.
✉email: mathieu.rougemaille@i2bc.paris-saclay.fr

Long non-coding RNAs (lncRNAs) are widespread in eukaryotic genomes and contribute to shape transcriptomes, acting in virtually all steps in gene expression, either as guides, scaffolds or decoys[1]. Often displaying tissue- or cell-specific expression profiles, lncRNAs are linked to human diseases, including cancer[2,3]. However, only a few cases of regulatory mechanisms have been described so far, highlighting the need to understand lncRNA function in gene expression and cell fate decisions.

The mitosis to meiosis switch is a key developmental transition that ensures the transmission of genetic information to offspring. In fission yeast *S. pombe*, meiosis occurs upon nutritional starvation and involves the stepwise induction of hundreds of genes[4,5]. A subset of meiosis-specific genes is also constitutively transcribed during mitosis but repressed at the post-transcriptional level to avoid untimely expression[6]. This RNA silencing pathway involves the YTH-family RNA-binding protein Mmi1, which selectively targets several mRNAs and lncRNAs for degradation by the nuclear exosome[6,7]. Mmi1 associates, via its YTH domain, with DSR (Determinant of Selective Removal) sequences enriched in repeats of the hexanucleotide motif UNAAAC[8–11].

Together with Mmi1, an intricate regulatory network of players suppresses meiotic mRNA expression during mitosis. Critical cofactors include the conserved Erh1 protein[12–14], components of the polyadenylation/termination machinery[15–18], and the multisubunit MTREC/NURS complex that physically bridges Mmi1 to the exosome[19–23]. Mmi1 also mediates selective termination of several lncRNAs to prevent their invasion into neighboring genes[11,24–27]. Adding another layer of regulation, Mmi1 cooperates with the heterochromatin and RNAi machineries to load chromatin repressive marks at a subset of meiotic genes[28–30].

During mitosis, Mmi1 colocalizes with MTREC and the exosome in scattered nuclear foci and mediates nuclear retention of meiotic mRNAs to prevent their export to the cytoplasm[23,31,32]. Upon meiosis onset, Mmi1 is lured by a ribonucleoprotein (RNP) complex, composed of the nucleocytoplasmic shuttling RNA-binding protein Mei2 and the DSR-containing lncRNA *meiRNA*, that accumulates in a single nuclear body overlapping the *meiRNA*-encoding *sme2 +* gene[6,33–36]. Both Mei2 and *meiRNA* are essential for the assembly of this structure and the completion of sexual differentiation. However, *meiRNA* overexpression during mitosis does not lead to meiotic mRNA accumulation and artificial targeting of Mei2 to the nucleus in *meiRNAΔ* cells partially restores meiosis[35,37], suggesting the existence of an additional mechanism responsible for Mmi1 inactivation.

Mitotic cells maintain low levels of *meiRNA* and Mei2. The former is targeted for exosome-dependent degradation by Mmi1 itself and the latter is phosphorylated by the Pat1 and Tor2 kinases, which facilitates its degradation by the ubiquitin and proteasome system[38–41]. Earlier studies further revealed that Mmi1 tightly associates with the evolutionarily conserved Ccr4-Not complex[12,42–45], yet its RNA deadenylation activity does not partake in meiotic RNA degradation in vivo. Rather, we previously showed that Mmi1 recruits Ccr4-Not to mediate ubiquitinylation and downregulation of a pool of its own inhibitor Mei2[45]. This regulatory circuit, which involves the E3 ubiquitin ligase subunit Mot2 of Ccr4-Not, is essential to maintain low levels of Mei2 in mitotic cells and hence preserve Mmi1 function in exosome-dependent degradation of meiotic RNAs, including its own decoy *meiRNA* (Fig. 1a). In turn, increased Mei2 levels in the absence of Mot2 lead to meiotic RNA accumulation due to Mmi1 inactivation. However, the precise molecular and cellular mechanisms by which Mmi1 and Mei2 reciprocally control their activities have remained elusive.

Guided by structural, genomic, and imaging approaches, we identify here a regulatory lncRNA, distinct from *meiRNA*, that scaffolds Mmi1 and Mei2 to exert their mutual control during mitosis. This lncRNA, termed *mamRNA*, localizes to a nuclear body enriched in Mmi1 and promotes meiosis even in the absence of *meiRNA*. Together, our results reveal the existence of a lncRNA-dependent and spatially confined regulation of Mmi1 and Mei2 activities.

## Results

**Mmi1 and Mei2 RNA-binding activities are required for their mutual control.** To investigate the mechanisms by which Mmi1 and Mei2 mediate their mutual control, we sought to determine the protein domains involved (Fig. 1b). We first assessed the impact of Mmi1 mutants defective for RNA-binding in Mei2 downregulation. Plasmid-borne Mmi1-*YTHΔ*, Mmi1-Y352F and Mmi1-Y466F mutants, previously shown to disrupt RNA-binding in vitro[9], were expressed in *mmi1Δ* cells carrying the deletion of *mei4 +*, since the absence of Mmi1 leads to severe viability defects due to ectopic expression of the meiosis-specific transcription factor Mei4[6]. In these mutants, the binding to DSR-containing meiotic mRNAs *ssm4 +* and *mcp5 +* and their subsequent degradation was abolished, while Mmi1 nuclear localization was preserved (Fig. 1c, Supplementary Fig. 1a, b). Importantly, these Mmi1 mutants exhibited elevated Mei2 levels, similar to *mmi1Δ* cells (Fig. 1d), and this was not due to a major increase in *mei2 +* gene expression (Supplementary Fig. 1a). We also detected higher mRNA and protein levels of Mmi1 mutants relative to the wild type, yet this was not sufficient to restore low Mei2 abundance (Fig. 1d, Supplementary Fig. 1a). Together, these results suggest that RNA-binding by Mmi1 YTH domain is necessary for efficient Mei2 downregulation.

In the absence of the E3 ubiquitin ligase Mot2, DSR-containing meiotic mRNAs accumulate in a Mei2-dependent manner as a consequence of Mmi1 inhibition[45]. To identify in turn the Mei2 domain(s) responsible for Mmi1 inactivation, we measured meiotic mRNA levels in *mot2Δ mei2Δ* cells expressing different plasmid-borne versions of Mei2. Interestingly, Mei2 mutants lacking the RRM3 domain (residues 592 to 750, hereafter called RRM3$^{Mei2}$) failed to accumulate meiotic mRNAs, suggesting that Mei2 RNA-binding activity is required to inhibit Mmi1 function (Supplementary Fig. 1c, d).

To gain insights into RNA recognition by RRM3$^{Mei2}$, we solved its crystal structure, which revealed a unique organization as the classical RRM fold is extended by the N-terminal helix α0 and by one helix and two strands at the C-terminal extremity (Supplementary Fig. 1e, Supplementary Table 1). According to the DALI server[46], RRM3$^{Mei2}$ also strongly resembles fly Sex-lethal (Sxl) RRM1 that recognizes uridine-rich stretches on RNAs[47] (Supplementary Fig. 1e). Using isothermal titration calorimetry (ITC) and size exclusion chromatography coupled to multi-angle laser light scattering (SEC-MALLS), we found that RRM3$^{Mei2}$ tightly interacts with a poly-U$_{15}$ in a 1:1 stoichiometry (Fig. 1e, Supplementary Fig. 1f,g Supplementary Table 2). Given the interaction of a guanine (G11) with Sxl F170 (Supplementary Fig. 1e), which corresponds to Mei2 F644 that is essential for Mei2 function in vivo[33], we also tested the influence of one (U$_7$GU$_7$) or two (U$_6$GGU$_7$) Gs in a poly-U$_{15}$ oligonucleotide and observed a considerable increase of Mei2 binding to these RNAs (Fig. 1e, Supplementary Fig. 1f, Supplementary Table 2).

We next solved the crystal structure of RRM3$^{Mei2}$ bound to the GCUUUUUGUUCG 12-mer (Fig. 1f, Supplementary Table 1). The RRM3$^{Mei2}$–RNA complex revealed that F644 and F634 side chains form π–π stacking interactions with U7 and G8, respectively (Fig. 1f, g, Supplementary Fig. 1k). As revealed by the PISA server[48], several Mei2 residues also form specific hydrogen

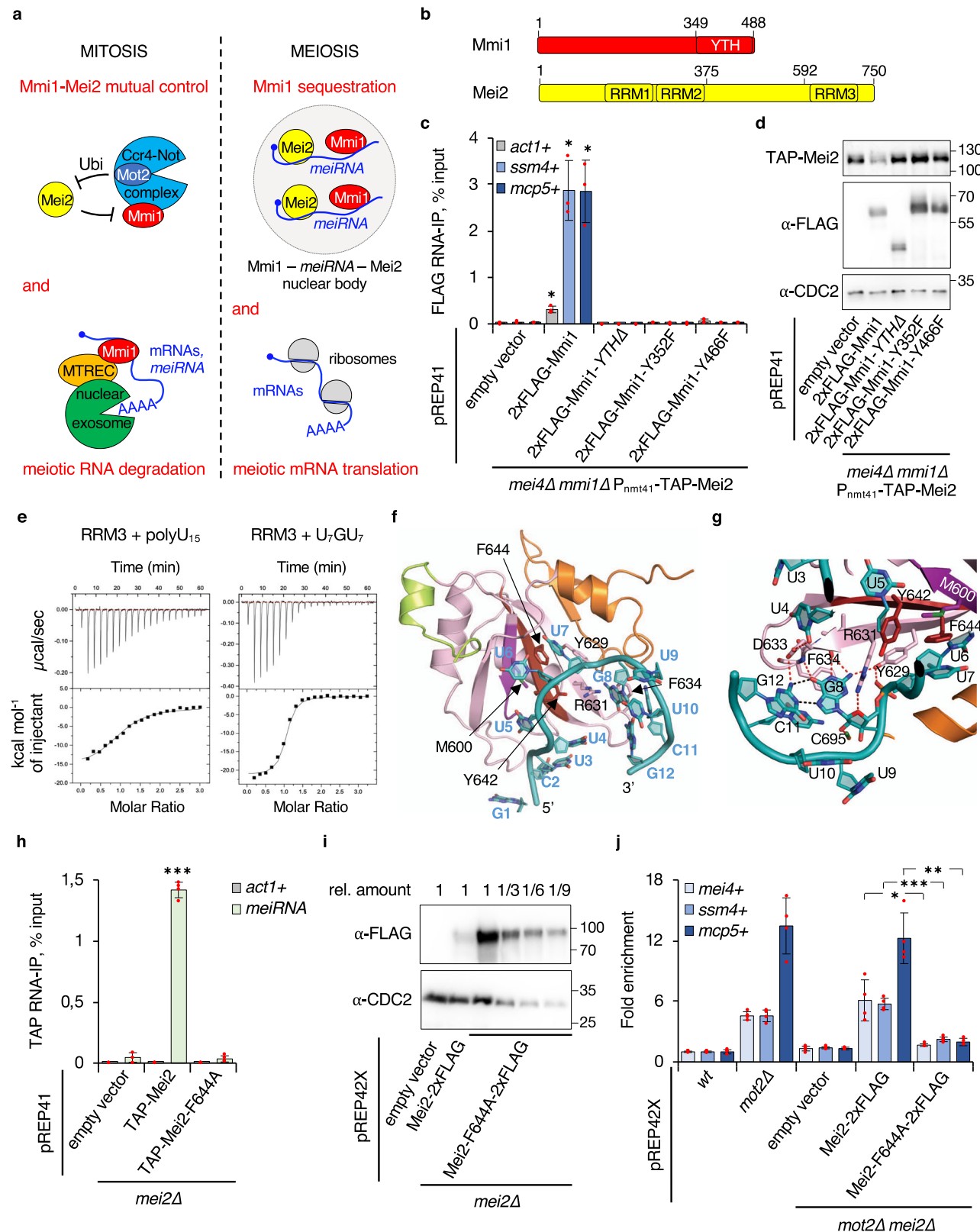

bonds with U4 to G8 (Supplementary Fig. 1h–k, Supplementary Table 3), rationalizing the stronger affinity for $U_7GU_7$ than for poly-$U_{15}$. However, the pocket involved in U7 recognition might also recognize a G, as observed in the RRM1$^{Sxl}$–RNA complex[47], or an A but not a C since the amino group attached to carbon 4 in

cytosine ring cannot form a hydrogen bond with the amino group from N680 main chain (Supplementary Fig. 1k). U4 binding pocket is solvent exposed and hence any nucleotide may fit in it. Collectively, our results suggest that RRM3$^{Mei2}$ recognizes UU(U/G/A)G motifs in RNA targets and provide a structural basis for

**Fig. 1 Mmi1 and Mei2 RNA-binding activities are required for their mutual control. a** Scheme summarizing the functional relationships between Mmi1 and Mei2 during mitosis and meiosis. **b** Domain organization of Mmi1 and Mei2 proteins. **c** Enrichments (% input; mean ± SD; $n = 3$ or 4) of $act1+$, $ssm4+$ and $mcp5+$ mRNAs upon pulldown of 2xFLAG-tagged *wt* or mutant Mmi1. Student's $t$ test (two-tailed) was used to calculate $p$-values. Between cells expressing pREP41-2xFLAG-Mmi1 and empty vector, $p = 0.01865$ ($act1+$); 0.0166 ($ssm4+$); 0.01819 ($mcp5+$) (0.05 > *>0.01). **d** Western blots showing total TAP-tagged Mei2 levels in cells of the indicated genetic backgrounds. Anti-FLAG and anti-CDC2 antibodies were used as Mmi1 expression and loading controls, respectively. **e** Upper panels: ITC data obtained by injecting poly-U$_{15}$ or U$_7$GU$_7$ RNAs to RRM3$^{Mei2}$. Lower panels: Fitting of the binding curves using a single binding site model. **f** Crystal structure of RRM3$^{Mei2}$ bound to the GCUUUUUGUUCG RNA (cyan). The classical RRM fold is shown in pink, with RNP1 and RNP2 motifs highlighted in brown and purple, respectively. The RNP1 F644 side chain is shown as sticks. The N- and C-terminal extensions are colored in green and orange, respectively. **g** Detailed representation of G8 binding mode. Hydrogen bonds involved in specificity for G8 are shown as red (protein–RNA interaction) or black (Hoogsteen based pairs between G8 and G12 bases) dashed lines. **h** Enrichments (% input; mean ± SD; $n = 3$ or 4) of $act1+$ mRNAs and *meiRNA* upon pulldown of TAP-tagged *wt* or mutant (Mei2-F644A) Mei2. Student's $t$ test (two-tailed) was used to calculate $p$-values. Between cells expressing pREP41-TAP-Mei2 and empty vector, $p = 4.32 \times 10^{-7}$ (*meiRNA*) (***<0.001). **i** Western blots showing total levels of 2xFLAG-tagged *wt* or mutant Mei2. Serial dilutions of the mutant extract are shown for comparison with wild type. Anti-CDC2 antibody was used as loading control. **j** RT-qPCR analyses of $mei4+$, $ssm4+$ and $mcp5+$ meiotic mRNA levels (mean ± SD; $n = 4$; normalized to $act1+$ and relative to *wt*) in cells of the indicated genetic backgrounds. Student's $t$ test (two-tailed) was used to calculate $p$-values. Between $mot2\Delta$ $mei2\Delta$ cells expressing pREP42X-Mei2-2xFLAG and pREP42X-Mei2-F644A-2xFLAG, $p = 0.02219$ ($mei4+$) (0.05 > *>0.01); 0.00035 ($ssm4+$) (***<0.001); 0.00338 ($mcp5+$) (0.01 > **>0.001). **c**, **h**, **j** Individual data points are represented by red circles.

the biological relevance of several Mei2 residues, including F644, whose substitution in alanine abolishes binding to *meiRNA*[33] (Fig. 1h, Supplementary Fig. 1f, Supplementary Table 2).

Importantly, Mei2$^{F644A}$ protein levels were roughly 10-fold more abundant than wild-type Mei2 (Fig. 1i). Although *mei2-F644A* mRNAs were also increased by 3-fold when compared to wild-type *mei2+* transcripts (Supplementary Fig. 1l), these data strongly suggest that Mei2 RNA-binding activity is required for its own downregulation. Further, expression of the Mei2-F644A mutant did not lead to the accumulation of meiotic mRNAs in *mot2Δ mei2Δ* cells (Fig. 1j), consistent with RRM3$^{Mei2}$ RNA-binding activity being also required for Mmi1 inactivation. These phenotypes did not result from an altered Mei2 cellular localization, as both the wild-type and mutant proteins distributed in the nucleus and the cytoplasm, consistent with former work[34,45] (Supplementary Fig. 1m).

**Identification of *mamRNA*, an Mmi1 and Mei2-scaffolding lncRNA.** The importance of Mmi1 and Mei2 RNA-binding activities raised the possibility that their mutual control could be RNA-mediated. We therefore determined the repertoire of transcripts simultaneously bound by both proteins through sequential immunoprecipitations followed by high-throughput sequencing of associated RNAs (seqRIP-seq) (Supplementary Fig. 2a, b). To enrich bound RNA species, we used mitotic cells overexpressing Mei2, since the levels of the latter are kept low in a wild-type strain. In these conditions, Mmi1 and Mei2 co-precipitated a large population of transcripts with various expression levels, including mRNAs, tRNAs, sn/snoRNAs, and lncRNAs (Fig. 2a, Supplementary Data 1). To narrow down the number of candidates, we intersected the 100 most enriched and 100 most abundant RNAs and identified 7 species falling in that category (Fig. 2a, red dots), including 5 sn/snoRNAs and 2 lncRNAs, *meiRNA* and *omt3*. Of note, the relatively high abundance of *meiRNA* in this experiment most likely results from Mei2 overexpression, which leads to the accumulation of Mmi1 RNA targets[45].

Visual inspection of sequencing data further revealed the strong enrichment of an unannotated feature located upstream of snoU14 (Supplementary Fig. 2c). RACE and Northern blot analyses demonstrated that this corresponded to an independent transcript, hereafter called *mamRNA* (for Mmi1 and Mei2-associated RNA), which accumulates as two non-adenylated isoforms and a pool of polyadenylated species (Fig. 2b, Supplementary Fig. 2d, e). *mamRNA* is produced by RNA PolII according to NET-seq data and is classified as a *bona fide*

lncRNA by CPC2 (Coding Potential Calculator 2)[49,50] (Supplementary Fig. 2f, g). Remarkably, *mamRNA* escapes degradation during mitosis (Fig. 2b), contrary to other Mmi1-bound meiotic transcripts including *meiRNA*. The presence of 3 Mmi1 binding sites (i.e. UNAAAC) as opposed to 25 in *meiRNA* may account for this difference (Fig. 2c), although a single motif was shown to be sufficient for RNA decay[7]. *mamRNA* also comprises 16 putative Mei2 motifs (i.e. UU(U/G/A)G) that are evenly distributed, akin to *omt3* and *meiRNA* (Fig. 2c). Since Mei2 associates with the latter in its 5' region[35], other sequence and/or structure elements likely specify its binding to RNA.

Using single molecule Fluorescence In Situ Hybridization (smFISH), we observed that *mamRNA* localizes to a prominent nuclear dot in mitotic cells, likely corresponding to its transcription site (Fig. 2d, e). A diffuse signal, distinct from DAPI staining, also indicated nucleolar localization, and scattered cytoplasmic spots further suggested that a pool of *mamRNA*, presumably the polyadenylated fraction, undergoes nucleocytoplasmic export (Fig. 2d, e). During meiotic prophase I, *mamRNA* accumulated to a lesser extent in scattered foci and partially redistributed to the cytoplasm (Supplementary Fig. 2h, i), suggesting the existence of mechanisms regulating its abundance and localization. This was in striking contrast to *meiRNA* that accumulates at its transcription site (i.e. the *sme2+* gene), luring Mmi1 and hence preventing meiotic mRNA degradation[35].

The possibility that an RNA bridges Mmi1 and Mei2 to ensure their reciprocal inhibition implies that its binding to both proteins should depend on their YTH and RRM3 domain, respectively (Fig. 1). Indeed, Mmi1 YTH mutants did not associate with the above-mentioned lncRNAs (i.e. *meiRNA*, *mamRNA*, and *omt3*), and Mei2-F644A binding was abolished or severely compromised (Figs. 1h, 2f, g). On the contrary, we did not observe significant RRM3-dependent enrichment for the other 5 sn/snoRNAs and we therefore excluded them for subsequent analyses (Supplementary Fig. 2j).

***mamRNA* mediates the Mmi1–Mei2 mutual control.** The model of a bridging RNA predicts that it should contribute to lower Mei2 abundance, similar to Mmi1 and Mot2, while promoting Mmi1 inactivation upon increased Mei2 levels (i.e. in *mot2Δ* cells). Strikingly, among the 3 lncRNAs, only *mamRNA* was required for Mei2 downregulation and this was independent from alterations in *mei2+* gene expression (Supplementary Fig. 3a,b). In addition, cells deleted for both Mot2 and *mamRNA* exhibited increased Mei2 levels similar to each single mutant, indicating

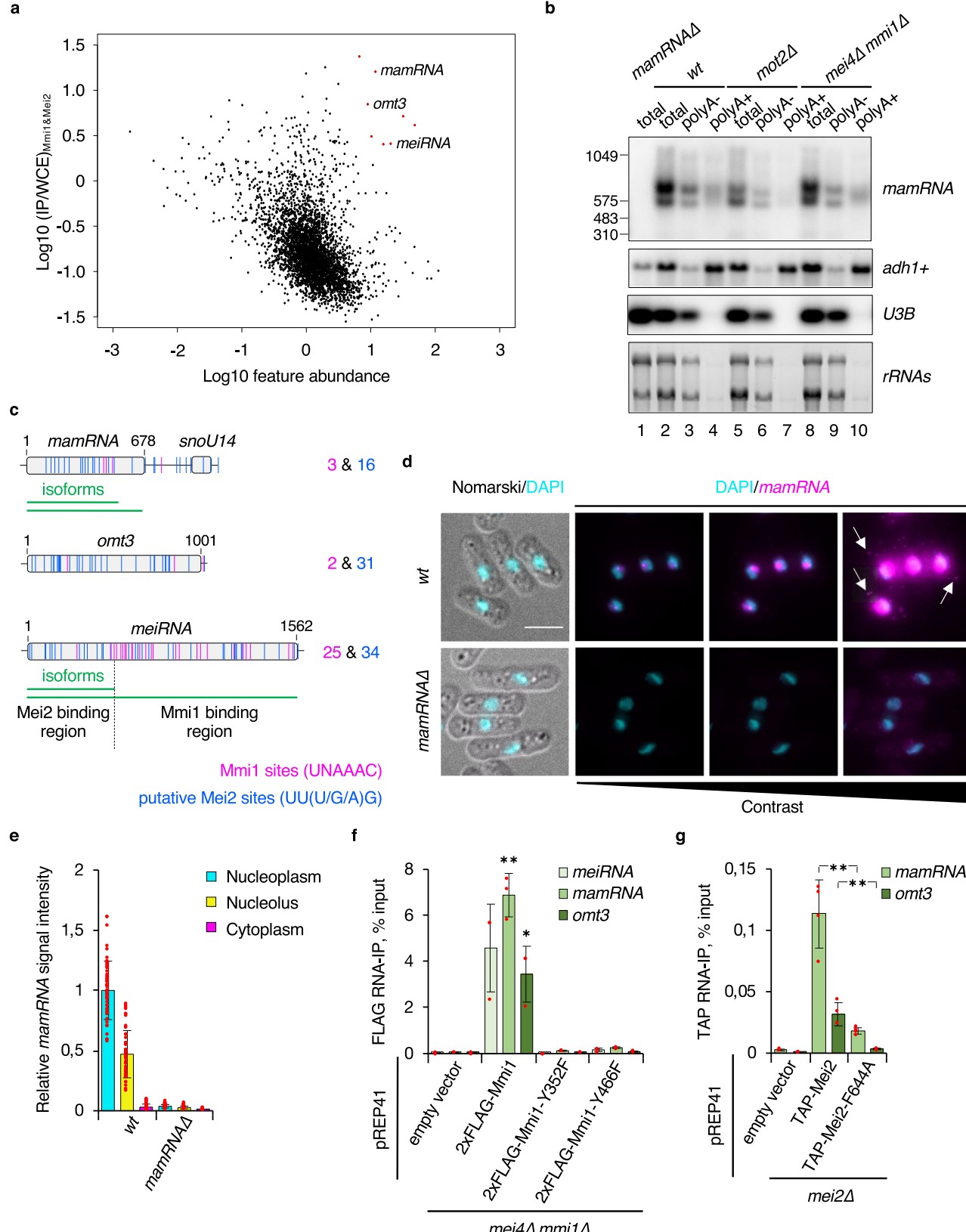

that they function in the same pathway (Fig. 3a). Ectopic expression of *mamRNA*, but not *snoU14*, from the *leu1 +* locus in otherwise *mamRNAΔ* cells, restores low levels of Mei2 (Fig. 3b), further revealing that it can function in *trans*.

To determine whether Mei2 dependent-inactivation of Mmi1 requires one of the three lncRNAs, we measured meiotic mRNA

levels in RT-qPCR assays. The absence of *mamRNA*, but not *meiRNA* or *omt3*, almost completely suppressed the accumulation of meiotic mRNAs in the *mot2Δ* background (Fig. 3c, Supplementary Fig. 3c). Transcriptome analyses further established a specific role for *mamRNA* in the accumulation of the entire set of Mmi1 targets in *mot2Δ* cells (Fig. 3d, Supplementary Fig. 3d,

**Fig. 2 Identification of *mamRNA*, an Mmi1 and Mei2-scaffolding lncRNA. a** SeqRIP-seq plot showing the enrichments (log10 IP/WCE) of transcripts pulled-down in the double-tagged strain of interest (*y* axis) as a function of their abundance (log10) (*x* axis) (*n* = 2). Red dots correspond to RNAs among the 100 most enriched and 100 most abundant features. Cells expressing TAP- or GFP-tagged Mmi1 and 3xFLAG-tagged Mei2 under the control of the mild nmt41 and strong nmt1 promoters, respectively, were used for immunoprecipitations (strain of interest: $P_{nmt41}$-TAP-Mmi1 $P_{nmt1}$-3xFLAG-Mei2; negative control: $P_{nmt41}$-GFP-Mmi1 $P_{nmt1}$-3xFLAG-Mei2). IP: Immunoprecipitate, WCE: Whole Cell Extract. **b** Northern blot showing *mamRNA* levels from total RNA samples and fractions enriched (polyA + ) or depleted (polyA-) for polyadenylated species, in the indicated genetic backgrounds. Enrichment and depletion of polyadenylated RNAs were verified by probing *adh1* + mRNAs and *snoU3B*, respectively. Ribosomal RNAs serve as a loading control. **c** Scheme depicting the chromosomal loci encoding the *mamRNA*, *omt3*, and *meiRNA* lncRNAs with the number and position of Mmi1 (magenta) and putative non-overlapping Mei2 (blue) binding motifs. *mamRNA* and *meiRNA* isoforms are shown in green below the corresponding loci. The regions of Mmi1 and Mei2 binding to *meiRNA* are indicated[35]. **d** Representative images of *mamRNA* detected by smFISH in *wt* and *mamRNAΔ* mitotic cells. DNA was stained with DAPI. Images are shown as maximum-intensity projections of Z-stacks. Distinct contrast adjustments are shown to visualize *mamRNA* subcellular localization. White arrows point to cytoplasmic spots. The white scale bar represents 5 μm. **e** *mamRNA* signal intensities (integrated densities relative to nucleoplasmic average; mean ± SD; *n* = 50 cells) quantified from **d** in the indicated areas. **f**, **g** Enrichments (% input; mean ± SD; *n* = 3 or 4) of *meiRNA*, *mamRNA* and *omt3* lncRNAs upon pulldown of *wt* or mutant 2xFLAG-tagged Mmi1 (**f**) and TAP-tagged Mei2 (**g**). Student's *t* test (two-tailed) was used to calculate *p*-values. Between cells expressing pREP41-2xFLAG-Mmi1 and pREP41-2xFLAG-Mmi1-Y352F or Y466F (**f**), *p* = 0.00641 or 0.00663 (*mamRNA*) (0.01 > \*\*>0.001); 0.03981 or 0.04065 (*omt3*) (0.05 > \*>0.01). Between cells expressing pREP41-TAP-Mei2 and pREP41-TAP-Mei2-F644A (**g**), *p* = 0.00608 (*mamRNA*); 0.00916 (*omt3*) (0.01 > \*\*>0.001). **e**, **f**, **g** Individual data points are represented by red circles.

Supplementary Data 2). Moreover, the sole deletion of *mamRNA* did not alter the expression of Mmi1 targets, despite increased Mei2 levels, thereby uncoupling Mmi1 inhibition from Mei2 accumulation (Fig. 3c, d). This supports the notion that *mamRNA* physically bridges Mmi1 and Mei2 to exert their mutual control in mitotic cells.

The ability of *mamRNA* to tune Mmi1 and Mei2 activities prompted us to assess its requirement for meiosis. Unlike *meiRNA*, we found that *mamRNA* is dispensable for sporulation (Supplementary Fig. 3e). Overexpression of an additional copy of *mamRNA* from the *leu1* + locus also failed to rescue sporulation defects in *meiRNAΔ* cells (Supplementary Fig. 3f). Increased *mamRNA* levels are therefore not sufficient to lure Mmi1 at meiosis onset, as opposed to the overexpression of DSR elements from meiotic mRNAs[6]. Crucially, however, *mamRNA* becomes critical in *meiRNAΔ* cells where artificial targeting of Mei2 to the nucleus partially restores meiosis[37] (Fig. 3e). This strongly suggests that, even in the absence of *meiRNA*, Mmi1 inhibition by *mamRNA* and Mei2 is sufficient to trigger meiosis.

**Subcellular localization of Mmi1, *mamRNA* and meiotic mRNAs**. We next localized *mamRNA* with respect to Mmi1 during mitosis. Strikingly, smFISH experiments in cells expressing GFP-tagged Mmi1 revealed a recurrent overlap between the prominent *mamRNA* nuclear signal (i.e. its likely transcription site) and one of the scattered Mmi1 foci in both wild-type and *mot2Δ* cells (Fig. 4a, Supplementary Fig. 4a, b), strongly suggesting that the *mamRNA*-dependent mutual control of Mmi1 and Mei2 is spatially confined. Our data further revealed that the localizations of Mmi1 and *mamRNA* were barely affected in the absence of Mot2 (Supplementary Fig. 4c, d).

To determine the mechanism underlying Mmi1 inactivation by *mamRNA* and Mei2 in *mot2Δ* cells, we investigated the localization of the *mcp5* + and *mei4* + meiotic mRNAs by smFISH, using cells deleted for Mmi1 and the MTREC subunit Red1 as controls for cytoplasmic and nuclear accumulation, respectively[32]. Compared to the wild-type strain, the number of *mcp5* + and *mei4* + foci increased substantially in *mot2Δ* cells, and co-deletion of *mamRNA* led to a marked decrease in the number of detected molecules (Fig. 4b, c, Supplementary Fig. 4e, f), in agreement with the molecular analysis (Fig. 3c, d). Deletion of Mmi1 or Red1 led to a major increase in *mcp5* + and *mei4* + spots, consistent with their direct role in meiotic mRNA degradation. Importantly, the majority of *mcp5* + and *mei4* + transcripts was restricted to the nucleus in the absence

of Mot2, similar to *red1Δ* cells but unlike those lacking Mmi1[32] (Fig. 4b, d, Supplementary Fig. 4e, g). This provides evidence that the nuclear retention activity of Mmi1 is preserved in the *mot2Δ* mutant. Together with RNA-IP experiments showing that Mmi1 still efficiently binds its RNA targets in this context (Fig. 4e), our data indicate that *mamRNA*-dependent inhibition of Mmi1 occurs downstream of target recognition and nuclear retention. *mamRNA* thus appears to function as a scaffold rather than a decoy in the inactivation of Mmi1 during mitosis.

## Discussion

In this study, we show that the RNA-binding activities of Mmi1 and Mei2 are crucial for their mutual control during mitosis and further identify a lncRNA, termed *mamRNA*, as a major regulatory and bridging partner for both proteins in the nucleus. Remarkably, one of the Mmi1-containing nuclear bodies localizes to the *mamRNA* transcription site, suggesting that the Mmi1–Mei2 mutual control is spatially confined (Fig. 4f).

The cooperation between *mamRNA* and Mmi1 to buffer Mei2 abundance on one side (i.e. in wt cells), and between *mamRNA* and Mei2 to inactivate Mmi1 on the other side (i.e. in *mot2Δ* cells), might be required to sustain robust mitotic growth or to prepare entry into meiosis, respectively. The *mamRNA*-dependent sensing of Mei2 levels may tip the scale towards a specific cell cycle as a function of environmental cues, thereby facilitating cell fate decisions. Mitotic cells exposed to temporary fluctuations in nutrient availability may also benefit from such a flexible system, shaping gene expression profiles when meiosis entry is too premature. The predominant role of *mamRNA* in fine-tuning Mmi1 and Mei2 activities during mitosis may thus reflect another facet of cell adaptability to environmental changes.

Prolonged nutritional starvation, which irreversibly commits cells to meiosis, triggers the induction of meiotic effectors, including Mei2 and *meiRNA*[51]. Coupled to the decrease in *mamRNA* levels, the build-up in Mei2 expression may, above a certain threshold, overwhelm its own downregulation and in turn promote Mmi1 inactivation. The resulting accumulation of DSR-containing transcripts, including *meiRNA*, would strengthen Mmi1 sequestration by enhanced Mei2 and *meiRNA* levels during meiotic prophase I[6,35]. Such a self-reinforcement system is likely to guarantee efficient Mmi1 inhibition, both lncRNP complexes (Mmi1–*mamRNA*–Mei2 and Mmi1–*meiRNA*–Mei2) acting in concert to ensure proper induction of the meiotic program. Crucially, *mamRNA* can take over *meiRNA* to induce meiosis,

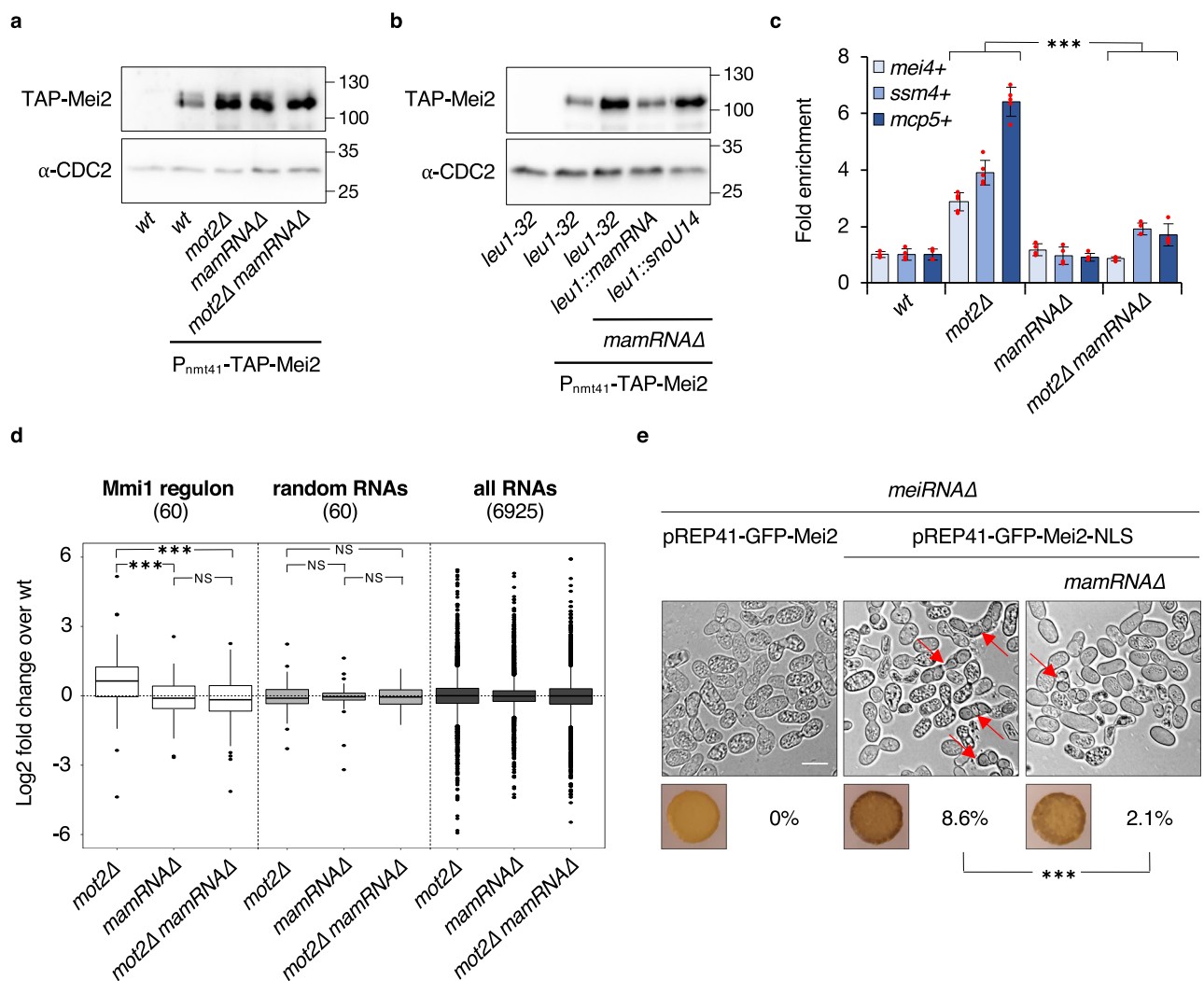

**Fig. 3 *mamRNA* mediates the Mmi1–Mei2 mutual control. a**, **b** Western blots showing total TAP-tagged Mei2 levels in cells of the indicated genetic backgrounds. Anti-CDC2 antibody was used as loading control. In **b**, *mamRNA* or *snoU14* were integrated at the *leu1*+ locus. **c** RT-qPCR analyses of *mei4* +, *ssm4*+ and *mcp5*+ meiotic mRNA levels in cells of the indicated genetic backgrounds (mean ± SD; $n = 4$ or 5; normalized to *act1*+ and relative to *wt*). Student's *t* test (two-tailed) was used to calculate *p*-values. Between *mot2Δ* and *mot2Δ mamRNAΔ* cells, $p = 0.00008$ (*mei4*+); $0.00011$ (*ssm4*+); $1.08 ×$ $10^{-6}$ (*mcp5*+) (***$<0.001$). Individual data points are represented by red circles. **d** Comparison of *wt*, *mot2Δ*, *mamRNAΔ* and *mot2Δ mamRNAΔ* transcriptomes by RNA-sequencing ($n = 2$). Box plots showing the fold enrichment (log2) of RNAs in mutants relative to *wt*. Left: analysis of 60 transcripts regulated by Mmi1 as defined in ref. [13] (cluster 1); Middle: analysis of 60 random transcripts; Right: analysis of all transcripts identified. Box center lines represent the median, box limits represent the upper and lower quartiles, whiskers define the 1.5x interquartile range and individual points correspond to outliers. An Anova test was used to calculate *p*-values. Between *mot2Δ* and *mamRNAΔ*, $p = 0.00046$ (***$<0.001$). Between *mot2Δ* and *mot2Δ mamRNAΔ*, $p = 0.00026$ (***$<0.001$). NS: not significant. **e** Mating/sporulation efficiencies of the indicated homothallic strains (% tetrads; $n_{meiRNAΔ\ pREP41-GFP-Mei2} =$ 699 cells; $n_{meiRNAΔ\ pREP41-GFP-Mei2-NLS} = 579$; $n_{meiRNAΔ\ mamRNAΔ\ pREP41-GFP-Mei2-NLS} = 674$), as determined by iodine staining and live cell imaging. Red arrows point to tetrads. A Fisher exact test (two-sided) was used to calculate the *p*-value. $p = 1.1517 × 10^{-7}$ (***$<0.001$). The white scale bar represents 10 μm.

suggesting that Mmi1 luring is not a prerequisite for the completion of sexual differentiation. *mamRNA* thus emerges as a critical regulator of Mmi1 activity upon meiosis onset.

Despite numerous transcriptomic studies in various growth conditions and genetic backgrounds, *mamRNA* escaped former characterization, probably due to its genomic location, immediately upstream the *snoU14*-encoding gene. Although its boundaries are clearly distinct from *snoU14*, *mamRNA* might be processed from a longer precursor transcript encompassing both features, as suggested by RNA PolII NET-seq profiles. Whether dedicated endoribonucleases mediate cleavage of such RNA species remains however to be addressed. Remarkably, a substantial fraction of *mamRNA* also localizes to the nucleolus, possibly reflecting additional functions in this compartment. In

addition, although specific lncRNAs can be translated into short, functional peptides[52], the requirement for Mmi1 and Mei2 RNA-binding activities in their mutual control rather support a direct role for the RNA molecule itself.

Despite its binding to Mmi1, *mamRNA* surprisingly escapes selective elimination as opposed to meiotic mRNAs and *meiRNA*. Whether this relates to its lower content in UNAAAC motifs, its polyadenylation status or both requires further investigation. The existence of structured regions and/or protein partners precluding exonucleolytic attack by the exosome, similar to sn/snoRNAs, may also provide a rationale for its stability in mitotic cells. As for Mei2 binding to *mamRNA*, additional elements such as longer sites and/or specific folds certainly provide more specificity than the sole UU(U/G/A)G motifs.

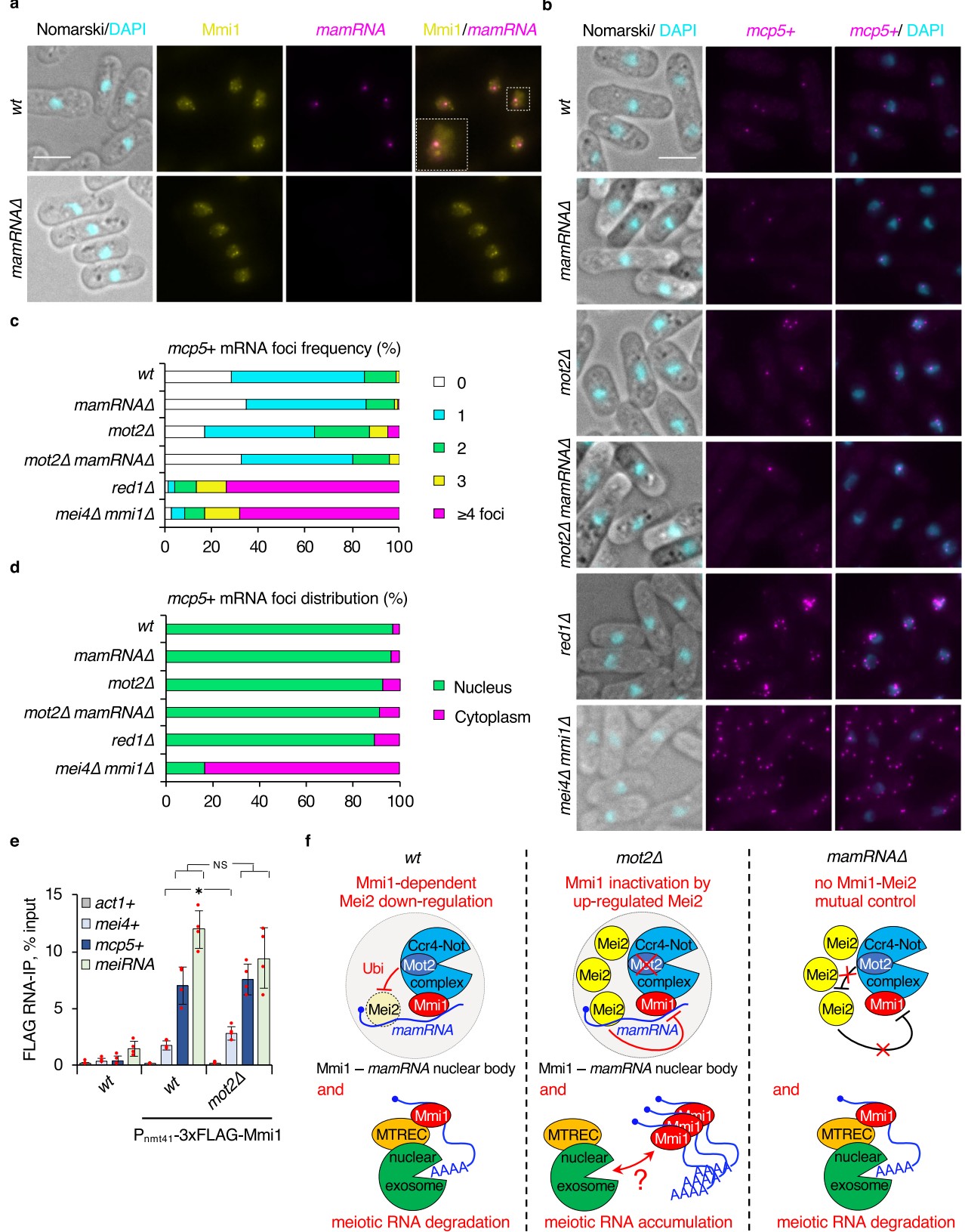

Akin to *meiRNA*[35], *mamRNA* transcription site constitutes a nucleation center for one Mmi1-containing nuclear body in mitotic cells. This might suggest that Mmi1 foci preferentially form at lncRNA transcription sites and that they do not solely reflect meiotic mRNA degradation centers as previously suggested. Interestingly, *omt3*, which we found to associate with

Mmi1 and Mei2, accumulates at its genomic locus to promote pairing of homologous chromosomes during meiosis, similar to *meiRNA*[53,54]. Whether *mamRNA* also contributes to this process remains an open question. Likewise, although the role of *omt3* in mitotic cells remains unclear, it is tempting to speculate that another Mmi1 body lies at this gene.

**Fig. 4 Subcellular localization of Mmi1, *mamRNA* and meiotic mRNAs. a**, **b** Representative images of *mamRNA* (**a**) or *mcp5* + mRNAs (**b**) detected by smFISH in cells of the indicated genetic backgrounds. DNA was stained with DAPI. In **a**, GFP-tagged Mmi1 was visualized in parallel. Images are shown as maximum-intensity projections of Z-stacks. White scale bars represent 5 µm. **c**, **d** Quantifications of smFISH analyses shown in **b**. **c** Distribution of *mcp5* + mRNA foci frequency per cell ($n_{wt}$ = 1003 cells; $n_{mamRNA\Delta}$ = 577; $n_{mot2\Delta}$ = 824; $n_{mot2\Delta mamRNA\Delta}$ = 240; $n_{red1\Delta}$ = 839; $n_{mei4\Delta mmi1\Delta}$ = 469). **d** Distribution of *mcp5* + mRNA foci localization ($n_{wt}$ = 872 foci; $n_{mamRNA\Delta}$ = 469; $n_{mot2\Delta}$ = 1166; $n_{mot2\Delta mamRNA\Delta}$ = 219; $n_{red1\Delta}$ = 939; $n_{mei4\Delta mmi1\Delta}$ = 1181). **e** Enrichments (% input; mean ± SD; $n$ = 4) of *act1* + , *mei4* + , *mcp5* + mRNAs and *meiRNA* upon pulldown of 2xFLAG-tagged Mmi1 in *wt* and *mot2Δ* cells. Student's *t* test (two-tailed) was used to calculate *p*-values. Between $P_{nm41}$-3xFLAG-Mmi1 and *mot2Δ* $P_{nm41}$-3xFLAG-Mmi1, $p$ = 0.02936 (*mei4* + ) (0.05 > *>0.01); 0.62718 (*mcp5* + ); 0.17136 (*meiRNA*). NS: not significant. Individual data points are represented by red circles. **f** Model depicting the role of *mamRNA* in the Mmi1–Mei2 mutual control in mitotic cells. In *wt* cells, Mmi1 binds to *mamRNA* to target Mei2 for downregulation by Mot2, which occurs in an Mmi1-containing nuclear body likely overlapping the *mamRNA* transcription site. This is required for efficient meiotic mRNA degradation by Mmi1, the Red1-containing MTREC complex and the nuclear exosome. In the absence of Mot2, increased Mei2 levels lead to meiotic mRNA accumulation in a *mamRNA*-dependent manner. Mmi1 inactivation occurs downstream of target recognition and nuclear retention. In *mamRNAΔ* cells, the Mmi1–Mei2 mutual control is abolished: Mei2 cannot be targeted to Mot2 but Mmi1 function in meiotic mRNA degradation is preserved.

In the absence of Mot2, accumulating Mei2 distributes throughout the cell[45], implying that only a fraction of the protein associates with *mamRNA* to in turn inactivate Mmi1. Available *mamRNA* binding sites may be limiting and/or the Mei2-*mamRNA* dissociation rate be slow. In this context, Mmi1 is still proficient for meiotic mRNA binding and nuclear retention, indicating that its inhibition occurs downstream of RNA recognition. Hence, Mmi1 interaction with its cofactor Erh1 is unlikely to be impaired as the latter is necessary for the former to bind meiotic transcripts[12,14]. It is however possible that Mei2-bound *mamRNA* partially impairs the association of Mmi1 with downstream effectors, such as components of the polyadenylation/termination machinery, MTREC and/or the exosome. An alternative but not mutually exclusive model predicts that increased Mei2 levels may modify the physicochemical properties of the Mmi1–*mamRNA* body, locally precluding efficient meiotic mRNA degradation.

The function of *mamRNA* in mediating the Mmi1–Mei2 mutual control profoundly revisits the current model of the mitosis to meiosis switch in fission yeast and defines a paradigm for lncRNA-based regulation of gene expression. Our study thus provides a conceptual framework to address the conservation of similar mechanisms in developmental transitions and diseases.

## Methods

**Strains, media, and plasmids.** The *S. pombe* strains used in this study are listed in Supplementary Table 4. Strains were generated by transformation following a lithium acetate-based method or by random spore analysis using complete medium (YE Broth, Formedium, #PMC0105) supplemented with appropriate antibiotics. All experiments were performed in 1X EMM-LEU-URA minimal medium (EMM-LEU-URA Broth, Formedium, #PMD0810) supplemented with 150 mg/L of either L-leucine (Sigma, #L8000) (EMM-URA), uracil (Sigma, #U750) (EMM-LEU) or both (EMM). To assess mating/sporulation efficiency, cells plated at 30 °C on ME (ME Broth, Formedium, #PCM0710) or EMM-LEU for 3 or 5 days, respectively, were exposed to iodine crystals (Sigma, #326143) which stain with dark color a starch-like compound in the spore wall.

The following plasmids were used for gene cloning/editing: pREP42X::URA4, pREP41::LEU2, pFA6a-kanMX6-P41nmt1-TAP, pFA6a-kanMX6-P41nmt1-3FLAG (Addgene plasmid # 19337; RRID:Addgene_19337)[55], pFA6a-natMX6-P3nmt1-3FLAG (Addgene plasmid # 19345; RRID:Addgene_19345)[55], pFA6a-kanMX6-P41nmt1-GFP (Addgene plasmid # 39290; RRID:Addgene_39290)[56], pFA6a-HyTkAX (Addgene plasmid # 73898; RRID:Addgene_73898)[57], pJK148-Pnmt1-GBP-mCherry(N)-leu1 + (NBRP plasmid FYP3968, pJQW#625; https://yeast.nig.ac.jp/yeast/fyPlasmidDetail.jsf?id=3968)[58].

**Cloning, expression, and purification of recombinant protein.** The DNA fragment encoding for RRM3[Mei2] domain (residues 579 to 750) was amplified by PCR with oligonucleotides oMG432 and oMG429 (Supplementary Table 5) and further cloned into pGEX-6P1 vector using Fast Digest *SmaI* and *NotI* restriction enzymes (Thermo Fisher Scientific, #FD0664 and #FD0595, respectively) to generate the plasmid pMG918 encoding for a GST-tag fused to the N-terminal extremity of the RRM3[Mei2] domain by a 3 C protease cleavage site. The RRM3[Mei2-F644A] point mutant was generated by one-step site-directed mutagenesis of pMG918 using oligonucleotides listed in Supplementary Table 5 to yield corresponding plasmids.

The RRM3[Mei2] proteins were overexpressed in BL21 (DE) Codon+ cells and in TBAI media (Terrific Broth Auto-Inducible media; Formedium, #AIMTB0260), containing Ampicillin (100 µg/mL; Sigma, #A9518) and Chloramphenicol (25 µg/mL; Applichem, #A7495). After OD$_{600}$ reached 0.6 at 37 °C, culture was transferred to 18 °C for 16 h. To overexpress selenomethionine labeled protein, 10 mL of overnight culture was used as inoculum for 1 L selenomethionine containing minimal media. Culture was grown at 37 °C until OD$_{600}$ reached 0.6 and then induced with 100 µM IPTG (Thermo Fisher Scientific, #R0392) and transferred to 18 °C for 16 h. Cells were harvested by centrifugation (4000 *g* for 45 min) and cell pellets were resuspended in 30 mL lysis buffer (20 mM Tris-HCl, pH 7.5, 200 mM NaCl, 5 mM 2-mercaptoethanol) in the presence of 100 µM PMSF. Cell lysis was performed by sonication on ice, followed by lysate clearance by centrifugation at 20000 *g* for 45 min. The supernatants were applied on GSH-sepharose 4B resin (VWR, # 17-0756-05) pre-equilibrated with lysis buffer. After extensive washing steps with lysis buffer and with a high salinity buffer (20 mM Tris-HCl pH 7.5, 2 M NaCl, 5 mM β-mercaptoethanol), the protein was eluted with elution buffer (20 mM Tris-HCl pH 7.5, 200 mM NaCl, 20 mM GSH, 5 mM β-mercaptoethanol). The eluted protein was next incubated overnight at 4 °C with GST-3C protease under dialysis conditions in lysis buffer and then passed again through GSH column to remove the GST-tag as well as the GST tagged 3 C protease. The unbound proteins were subjected to an Heparin Sepharose 6 Fast flow column (GE Healthcare Biosciences, # 17-0998-01) and eluted using a linear gradient ranging from 100% Buffer A (50 mM Tris HCl, pH 7.5, 50 mM NaCl, 5 mM 2-mercaptoethanol) to 100% Buffer B (50 mM Tris HCl, pH 7.5, 1 M NaCl, 5 mM 2-mercaptoethanol). The protein eluted from Heparin column was further purified using a HiLoad 16/600 Superdex 75 column (GE Healthcare Biosciences, # 28-9893-33) pre-equilibrated with lysis buffer on an ÄKTA Purifier system (GE Healthcare Biosciences).

**Crystallization, data collection, and structure determination.** Crystals of the native or SeMet-labeled RRM3[Mei2] proteins in the absence of RNA were obtained by mixing 1 µL of concentrated protein (20 mg/mL in the lysis buffer) with an equal volume of crystallization solution (1 M ammonium sulfate, 0.1 M Bis-Tris pH 5.5; 1% PEG 3350) at 4 °C.

Prior to crystallization assays with RNA, 1 mM of RRM3[Mei2] (20 mg/ml) was incubated with 1.2 molar excess of the GCUUUUUGUUCG oligonucleotide (purchased from Dharmacon). Then, 1 µL of this solution was mixed with an equal volume of crystallization condition composed of 0.2 M NaCl, 0.1 M Na/K phosphate pH 6.5, 25% w/v PEG 1000 at 4 °C.

For data collection, all crystals were quick-soaked in cryo-protectant solutions containing 15% (v/v) and then 30% glycerol in corresponding crystallization solutions and flash-frozen in liquid nitrogen to collect the data set. Datasets were collected on Proxima-1 (WT and Se-Met labeled apo-structure) and Proxima-2A (RNA-bound structure) beamlines at SOLEIL Synchrotron (Saint-Aubin, France). Statistics for data processing are summarized in Supplementary Table 1.

The RRM3[Mei2] apo-structure was solved by Se-SAD using data collected at a wavelength close to the absorption edge of selenium. Se sites were located by SHELXD[59] while experimental phasing and automated model building was performed with the AutoSol procedure[60] implemented in the Phenix suite. A first model was obtained by iterative cycles of building and refinement performed using COOT and BUSTER programs, respectively. The structure was further refined at high resolution using the 1.9 Å resolution native dataset. The R and R$_{free}$ of the final structure are 17.1% and 19.4%, respectively. In the final structure, RRM3[Mei2] residues 580–726 and 580–725 are visible for promoters A and C, respectively. In addition, 9 molecules of Ethylene glycol from cryo-protectant, 3 molecules of PEG and 3 sulfate ions as well as 187 water molecules were modeled in the electron density maps.

The structure of the RRM3[Mei2] - RNA complex was solved by molecular replacement using the coordinates of the apo-structure as template with the PHASER program[61]. The final structure was obtained at 2.65 Å resolution after multiple rounds of model building and refinement as for the apo-structure. The final model had a R and R$_{free}$ of 22.3% and 29.8%. The final structure contains residues 580–727 from RRM3[Mei2], the 12 nucleotides from the RNA fragment, and 30 water molecules.

Statistics for structure refinement are summarized in Supplementary Table 1.

**Isothermal titration calorimetry**. Prior to ITC analysis, protein and RNA samples were dialyzed against the following buffer: 20 mM Tris HCl pH 7.5, 200 mM NaCl, 5 mM 2-mercaptoethanol. The interaction between RRM3$^{Mei2}$ and different RNA fragments (Dharmacon) was characterized using an iTC$_{200}$ machine at 20 °C. In all ITC experiments, 200 μL of protein was titrated by several injections of 2 μL of RNA at intervals of 180 s and stirring speed of 700 rpm. The concentrations of the reagents in the cell (proteins) and syringe (RNAs) are indicated in Supplementary Table 2. A theoretical curve assuming a one-binding site model calculated using Origin Software (MicroCal Inc.) gave the best fit to the experimental data. As SEC-MALLS analysis revealed a 1:1 stoichiometry, the $N$ value was fixed to 1 during fitting.

**Size exclusion chromatography-multi angle laser light scattering (SEC-MALLS)**. For each experiment, a 100 μL (1 mg/mL) sample was injected at a flow rate of 0.75 mL/min on a Superdex$^{TM}$ 200 Increase 10/300 GL column (GE-Healthcare Biosciences, # 28-9909-44) in the same buffer as for ITC experiments (20 mM Tris HCl pH 7.5, 200 mM NaCl, 5 mM 2-mercaptoethanol). Elution was followed by a UV-Visible spectrophotometer, a RID-20A refractive index detector (Shimadzu), a MiniDawn TREOS detector (Wyatt Technology). The data were collected and processed with the program ASTRA 6.1 (Wyatt Technology). M$_w$ was directly calculated from the absolute light scattering measurements using a dn/dc value of 0.183.

**Total protein analyses**. Total proteins were extracted from cell pellets corresponding to 2 to 5 ODs, as previously described[45]. Briefly, cell lysis was performed on ice using 0.3 M NaOH and 1% beta-mercaptoethanol prior to protein precipitation with trichloroacetic acid (TCA) (7% final). Following full speed centrifugation, pellets were resuspended in HU loading buffer and heat-denatured at 70 °C. Soluble fractions were recovered, and samples were analyzed by standard immunoblotting procedures using 1:3000 peroxydase-conjugated antiperoxydase (PAP, to detect protein-A-tagged proteins) (Sigma, #P1291, RRID:AB_1079562), 1:3000 monoclonal anti-FLAG antibody (Sigma, #F3165, RRID:AB_259529), 1:3000 anti-CDC2 antibody (Abcam, #ab5467, RRID:AB_2074778), 1:1000 anti-GFP antibody (Roche, # 11814460001, RRID: AB_390913), 1:500 anti-CBP antibody (Millipore, #07-482, RRID:AB_310653), 1:5000 goat anti-mouse IgG-HRP (Santa Cruz Biotechnology, #sc-2005, RRID:AB_631736) and 1:5000 goat anti-rabbit IgG-HRP (Santa Cruz Biotechnology, #sc-2004, RRID:AB_631746). Detection was done with SuperSignal West Pico Chemiluminescent Substrate (ThermoFisher Scientific, #34080), ECL Select reagent (GE Healthcare, #RPN2235), and a Vilber Lourmat Fusion Fx7 imager.

**Total RNA extraction**. Total RNAs were extracted from cell pellets using hot acid phenol, as described in ref. [45]. Samples were treated with DNAse (Ambion, #AM1906) and concentrations measured with a Nanodrop.

**Polyadenylated RNA purification from total RNA**. 50 μl of DYNAL Dynabeads Oligo(dT)$_{25}$ (ThermoFisher Scientific, #61002) were used to purify polyadenylated transcripts from 20 μg total RNAs. Purification was performed according to the manufacturer's instructions except that the first step flow-through was incubated a second time with the same beads. Eluted fractions corresponded to the "polyA +" fraction and the flow-through was recovered as the "polyA-" fraction.

**RT-qPCR**. 2 μg of DNAse-treated RNAs were denatured at 65 °C for 5 min in the presence of strand-specific primers. Reactions were carried out with 100 units Maxima Reverse Transcriptase (ThermoFisher Scientific, #EP0743) at 50 °C for 30 min. The enzyme was then denatured at 85 °C for 5 min, and reactions diluted to 1:10 ratio. Each experiment included negative controls without Reverse Transcriptase. Samples were analyzed by qPCR with SYBR Green and a LightCycler LC480 apparatus (Roche). Quantification was performed using the ΔCt method. Oligonucleotides used in qPCR reactions are listed in Supplementary Table 5.

**Northern blotting**. 3 to 5 μg RNAs were separated on a 1.5% agarose gel and transferred overnight by capillarity on a nylon membrane (GE Healthcare, #RPN203B) in SSC 10X Buffer. RNAs were then UV-crosslinked to the membrane using a Stratalinker apparatus. Generation and incubation of the membrane with RNA dig-labeled probes was performed using the Dig-Northern-Starter Kit (Roche, #12039672910), following manufacturer's instructions. Membranes were washed twice in 2X SSC 0.1% SDS and once in 1X SSC 0.1% SDS, for 10 min at 65 °C. Revelation was done according to the kit instructions using a Vilber Lourmat Fusion Fx7 detection device. Oligonucleotides used to generate DNA templates for RNA probes are listed in Supplementary Table 5.

**5′ and 3′ RACE analyses**. The 5′/3′ RACE Kit, 2$^{nd}$ Generation (Roche, #3353621001), was used to determine the 5′ and 3′ ends of *mamRNA* and *snoU14*, following manufacturer's instructions. An RNA polyadenylation step was included to map the 3′ ends of non-adenylated species. Oligonucleotides used to amplify and clone RACE products are listed in Supplementary Table 5.

**RNAseH assays**. 10 μg of DNAse-treated RNAs were incubated in 10X RNAseH Buffer at 65 °C for 15 min and 30 °C for 10 min in the presence of 25 μM primer. RNAseH (New England Biolabs, #M0297S) was then added to the reaction and samples were incubated at 30 °C for 45 min. Following phenol-chloroform extraction and ethanol precipitation, digested RNAs were resuspended in a loading buffer containing formamide, bromophenol, and cyanol blue. Half of reactions were denatured at 85 °C for 5 min and loaded on a 2% agarose gel prior to Northern blotting. Undigested controls, without primer in the reaction, were processed in parallel. Oligonucleotides used for RNaseH cleavage are listed in Supplementary Table 5.

**Native RNA-Immunoprecipitation**. 40 to 50 ODs of cells were grown to mid-log phase at 30 °C in EMM or EM-LEU and harvested by centrifugation. Cell pellets were washed in 1X PBS and resuspended in 2 ml lysis buffer (6 mM Na$_2$HPO$_4$, 4 mM NaH$_2$PO$_4$, 150 mM NaC$_2$H$_3$O$_2$, 5 mM MgC$_2$H$_3$O$_2$, 0.25% NP-40, 2 mM EDTA, 1 mM EGTA, 5% glycerol, 2 mM benzamidine, 1X complete EDTA-free protease inhibitor cocktail (ThermoFisher Scientific, #A32955) and 80 U RNaseOUT Ribonuclease inhibitor (Invitrogen, #10777-019) to make 'pop-corn'. Lysis was performed using a Ball Mill (Retsch, MM400) for 15 min at a 10 Hz frequency. Extracts were cleared by centrifugation before precipitation with 40 μL pre-washed anti-FLAG M2 affinity gel (Sigma, #A2220) for 2 hr at 4 °C. Beads were then washed twice with IPP150 (10 mM Tris pH 8, 150 mM NaCl, 0.1% NP-40). Total and immunoprecipitated RNAs were extracted with phenol: chloroform 5:1 pH 4.7 (Sigma, #P1944) and precipitated with ethanol. RNA samples were treated with DNase (Ambion, #AM1906) prior to RT-qPCR analyses as mentioned above.

Note that this procedure was followed in all experiments involving Mmi1 as a bait protein (Figs. 1c, 2f, 4e).

**Formaldehyde RNA-immunoprecipitation**. 40 to 50 ODs of cells were grown to mid-log phase at 30 °C in EMM-LEU and cross-linked with 0.2% formaldehyde for 20 min. Cells were harvested by centrifugation following quenching with 250 mM glycine for 5 min at room temperature. Cell pellets were washed in 1X PBS and resuspended in 2 ml RIPA buffer (50 mM Tris-HCl pH 8, 150 mM NaCl, 1% NP-40, 0.5% sodium deoxycholate, 0.1% SDS, 2 mM EDTA, 2 mM benzamidine, 1X complete EDTA-free protease inhibitor cocktail, and 80 U RNaseOUT Ribonuclease inhibitor to make "pop-corn". Lysis was performed using a Ball Mill (Retsch, MM400) for 15 min at 15 Hz frequency. Extracts were cleared by centrifugation before immunoprecipitation with 1 mg of pre-washed rabbit IgG-conjugated M-270 Epoxy Dynabeads (Invitrogen, #14311D) for 1 h at 4 °C. Beads were then washed once with low salt buffer (10 mM Tris-HCl pH 7.5, 150 mM NaCl, 0.5% Triton X-100), twice with high salt buffer (10 mM Tris-HCl pH 7.5, 1 M NaCl, 0.5% Triton X-100), and once again with low salt buffer for 10 min at room temperature. Samples were decrosslinked at 70 °C for 45 min in the presence of reverse buffer (10 mM Tris-HCl pH 6.8, 5 mM EDTA, 10 mM DTT, 1% SDS) and treated with proteinase K (Euromedex, #09-0911) for 30 min at 37 °C. Total and immunoprecipitated RNAs were next extracted with phenol:chloroform 5:1 pH 4.7 (Sigma, #P1944), precipitated with ethanol, and treated with DNase (Ambion, #AM1906) prior to RT-qPCR analyses.

Note that this procedure was systematically used upon Mei2 pulldown since the protein is highly susceptible to proteolysis under native conditions (Figs. 1h, 2g, Supplementary Fig. 2j).

**Sequential RNA-immunoprecipitation and high-throughput sequencing (seq-RIP-seq)**. 250 ODs of cells were grown to mid-log phase at 30 °C in EMM and cross-linked with 0.2% formaldehyde for 20 min. Cells were harvested by centrifugation following quenching with 250 mM glycine for 5 min at room temperature. Cell pellets were washed in 1X PBS and resuspended in 2 ml RIPA buffer (50 mM Tris-HCl pH 8, 150 mM NaCl, 1% NP-40, 0.5% sodium deoxycholate, 0.1% SDS, 2 mM EDTA, 2 mM benzamidine, 1X complete EDTA-free protease inhibitor cocktail and 80 U RNaseOUT Ribonuclease inhibitor to make "pop-corn". Lysis was performed using a Ball Mill (Retsch, MM400) for 15 min at 15 Hz frequency. Extracts were cleared by centrifugation before immunoprecipitation with 5 mg of pre-washed rabbit IgG-conjugated M-270 Epoxy Dynabeads (Invitrogen, #14311D) for 1 h at 4 °C. Beads were then washed once with low salt buffer (10 mM Tris-HCl pH 7.5, 150 mM NaCl, 0.5% Triton X-100), twice with high salt buffer (10 mM Tris-HCl pH 7.5, 1 M NaCl, 0.5% Triton X-100) and once again with low salt buffer for 10 min at room temperature. Samples were eluted with TEV protease (Invitrogen, #12575015) for 2 h at 16 °C with occasional shacking. Eluates were next used for a second immunoprecipitation step with 40 μL pre-washed anti-FLAG M2 affinity gel (Sigma, #A2220) for 2 h at 4 °C. Beads were washed as described above. Total and immunoprecipitated RNAs were extracted with phenol: chloroform 5:1 pH4.7 (Sigma, #P1944) and precipitated with ethanol. RNA samples were treated with DNase (Ambion, #AM1906) prior to analysis by Illumina sequencing.

Note that several Mmi1-targeted DSR-containing meiotic mRNAs were pulled-down, most likely as the consequence of Mei2 ectopic binding due to protein overexpression. The absence of meiotic mRNAs bound to endogenous Mei2 upon meiosis onset supports this notion[36].

**RNA-sequencing of total or immunoprecipitated RNAs**. Ribosomal RNA depletion was performed on whole-cell extract and total RNAs using RiboZero gold (Illumina) according to the manufacturer's instructions. NextSeq 500/550 Mid Output Kit v2 was used to prepare RNAseq libraries and paired-end 75 sequencing was performed on a NextSeq instrument. Reads were demultiplexed (bcl2fastq2-2.18.12) and adapters were trimmed (Cutadapt 1.15). Sequencing data were then uploaded to the Galaxy web platform, and public server (usegalaxy.org) was used for data analysis[62]. Reads were aligned to the S. pombe genome using Bowtie2, and MulticovBed generated the read count files. Processed data were analyzed using R and differential analysis of total RNA-sequencing was done using Sartools R Pipeline[63].

**Live cell microscopy**. Exponentially growing cells cultured in minimal medium (EMM-LEU) were stained with Hoescht 33342 (ThermoFisher Scientific, #62249) prior to room temperature imaging with a motorized Leica DMIRE2 inverted fluorescence microscope equipped with a 100X oil immersion objective (Leica), a charge-coupled device (CCD) camera (Cool SNAP HQ²; Photometrics), and the MetaMorph 6.1 acquisition software. DNA and GFP-tagged proteins were visualized using DAPI (Semrock DAPI 1160b-LSC-Zero) and GFP (Semrock FITC 2024b-LSC-Zero) filter sets. Images were processed in ImageJ (NIH).

**Single molecule fluorescence in situ hybridization (smFISH)**. Quasar 570 or 670-labeled mamRNA, meiRNA, mei4 + and mcp5 + mRNA probes were designed using Stellaris Probe Designer tool (Supplementary Table 6) and synthesized by Biosearch Technologies. Single molecule RNA Fluorescence In-Situ Hybridization (smFISH) was performed according to the manufacturer's protocol (Biosearch Technologies) with minor modifications.

All experiments were performed as described in ref. [14]. Briefly, vegetative cells were grown to mid-log phase at 30 °C in EMM or plated on ME medium prior to incubation for 40 h at 30 °C. Cells were then resuspended in 1X PBS containing 3.7% formaldehyde, treated with Zymolyase 100 T for cell wall digestion and permeabilized in 70% ethanol prior to overnight incubation with relevant probes. Stellaris RNA FISH hybridization and wash buffers were obtained from Biosearch Technologies. DAPI stained cells were resuspended in Vectashield antifade mounting medium (Vector laboratories) and imaged using a Leica DM6000B microscope with a 100X, NA 1.4 (HCX Plan-Apo) oil immersion objective equipped with a piezo-electric motor (LVDT; Physik Instrument) mounted underneath the objective lens, and a CCD camera (CoolSNAP HQ; Photometrics). Maximum intensity-projection of optical Z-sections (0.2 μm, 25 planes) was performed in ImageJ (NIH).

To compare mamRNA levels in different cell compartments, nucleolar, nuclear, and cytoplasmic signals were quantified based on the manual delineation of the corresponding areas and the measure of the associated signal intensities. In all cases, background levels from regions of the same surface were subtracted to measured values.

**Statistics and reproducibility**. All experiments comprising RT-qPCR assays (total RNA levels, RNA-IPs) were repeated at least three times, involving at least two independent biological duplicates. qPCR measurements were statistically compared using two-tailed t-tests with the following p-value cut-offs for significance: 0.05 > *>0.01; 0.01 > **>0.001; ***<0.001.

All representative images underlying Western and Northern blotting, live cell microscopy, and smFISH analyses were obtained from experiments repeated at least two or three times involving independent biological replicates.

The measure of RNA signal intensities (integrated density) in smFISH analyses was performed on at least 50 cells using ImageJ (NIH).

**Reporting summary**. Further information on research design is available in the Nature Research Reporting Summary linked to this article.

## Data availability

The atomic coordinates and structures of apo- and RNA-bound RRM3$^{Mei2}$ have been deposited to the Protein Data Bank with the accession codes 6YYL and 6YYM. Genomic datasets have been deposited to the Gene Expression Omnibus with the accession numbers GSE152159 and GSE152383. All data are available from the authors upon reasonable request. Source data are provided with this paper.

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

## Acknowledgements

We are grateful to Jürg Bahler, Nathalie Bonnefoy, Jean-Paul Javerzat, Antonin Morillon, Taro Nakamura, Eishi Noguchi, John Pringle, Kazunori Tomita, and André Verdel for gift of strains and plasmids. We thank Mireille Bétermier, Alain Jacquier, Domenico Libri, and Antonin Morillon for critical reading of the manuscript. We acknowledge SOLEIL for provision of synchrotron radiation facilities and technical assistance, as well as the High-throughput sequencing facility of I2BC for its sequencing and bioinformatics expertise. This work was supported by the Ecole polytechnique, the Centre National pour la Recherche Scientifique and the Agence Nationale de la Recherche (ANR-16-CE11-0003 to M.G., ANR-18-CE12-0003 to B.P., ANR-16-CE12-0031 to M.R.). D.H. was supported by a PhD fellowship from the French Ministry of Research and a mini post-doctoral fellowship from Ecole polytechnique.

## Author contributions

V.A., A.N., and D.H. conceptualized and designed the work, acquired, analyzed and interpreted data, and revised the manuscript. S.A. and A.M. acquired, analyzed and interpreted data, and revised the manuscript. M.G., B.P., and M.R. conceptualized and designed the work, analyzed and interpreted data, and drafted and revised the manuscript. M.R. supervised the work.

## Competing interests

The authors declare no competing interests.
