## [Peer Review File · Nature Communications]

REVIEWER COMMENTS

Reviewer #1 (Remarks to the Author):

This is a very interesting paper in which the authors identify a lncRNA that is important for switching from mitosis to meiosis. The authors identify the mamRNA that binds two crucial factors, Mmi1 and Mei2, which form a regulatory network.

Mmi1 is an RNA-binding protein associated with two major RNA degradation complexes, MTREC/Exosome, and the CCR4-NOT deadenylation complex. Mei2 was previously shown to bind another ncRNA, meiRNA, and this meiRNA-Mei2 RNP acts as a sponge to sequester Mmi1. The meiRNA, Mmi1 and Mei2 together form a regulatory circuit that controls the Mmi1 and Mei2 levels and thereby regulates RNA degradation. With the newly identified mamRNA, the authors add another regulator to this circuit.

The experiments are overall of good quality and also the figures are quite clear.

In general, it would be helpful for the reader if the experiments are introduced a little bit more detailed. For example, it is not clear why the authors use a mei4 deletion strain throughout their work. Please explain this. Similarly, the Mmi1 mutants defective for RNA-binding are introduced without explanation of how the residues for mutation were chosen (also see below).

More specific:

The center of this paper is the identification of the mamRNA, which contains predicted binding sites for Mmi1 and Mei2.

As this RNA is the most critical molecule of this paper, it deserves more attention. Conceptually, the mamRNA and the meiRNA are similar because both can bind Mmi1 and Mei2, but the mamRNA can obviously do something the meiRNA cannot. The mechanistic question is: Which features of the mamRNA make it a stronger regulator than the meiRNA.

As the paper is in several parts mechanistic/atomistic, I think it would be beneficial to

-Show the mamRNA sequence schematically (secondary structure/folding prediction) and highlight the positions of Mei2 (UU(U/G)G) and Mmi1 (UNAAAC) binding motifs in this scheme.

How does the mamRNA compare to the meiRNA, both predicted folding and the presence/number of Mei2 and Mmi1 binding motifs?

Would this maybe help to explain why one RNA escapes degradation during mitosis in contrast to meiRNA? Is it merely more structured or stable?

The authors show at the very beginning that the mutations/deletions of RNA-binding domains of Mmi1 and Mei2 abrogate the mutual Mmi1/Meo2 regulatory system. Later they show that the deletion of the mamRNA phenocopies the mot2 or mei4/mmi1 deletion strains for Mei2-levels. As the binding motif for the Mmi1 YTH motif is known, and the authors here identify the Mei2 binding motif, it would have been fantastic (and even more mechanistic) if the authors would have deleted or mutated these motifs within the mamRNA (or perform rescue experiments with different mamRNA constructs).

How did the authors find the similarity to Sxl RRM1? Does this come from a PDB wide comparison e.g., Dali search?

Please show the ITC titrations. A table with the thermodynamic data does not allow to judge the quality of the experiment. It is also not clear how the data were fitted, as the stoichiometry n is not indicated. Was n fixed to 1, after determination of the 1:1 stoichiometry from MALLS? Please also list the error limits from the estimated fit.

The structural data look very solid and, together with the ITC data, allow to suggest an RNA-binding motif recognized by the Mei2 RRM3. However, I am not entirely convinced about the UU(U/G)G motif. Changing one internal U to a G increased the affinity by a factor of 7. Replacing an upstream U by a G

then again reduces the affinity by a factor of 2. This clearly shows how the preference of UUUG > UUGG > UUUU. The nucleotide at position seven does not form many hydrogens bonds, so how do the authors exclude the possibility of the presence of A or C at position 7? Could the motif also be UUNG?

Line 115: 'recognition of a G11 by Slx1 F170' – I agree that the F can participate pi-stacking with the U base, but base-specific recognition is usually realized by reading out hydrogen bonding patterns (e.g., see S1g). Please change 'recognize' to 'interact'.

Minor points

Supplementary Figure 1b - Hoescht should read Hoechst

Supplementary Figure 1f – the unit for the refractive index is not mAU (Milli absorption units). Do the authors mean arbitrary units? Please specify. It would help note the mass of the U15 RNA so that the theoretical mass of the complex can be compared to the measured.

Line 113 - MALLS not explained, but SEC is

Supplementary Table 1: please divide the one table as these are three different things and add the appropriate descriptions required to understand the table.

-Data collection and refinement (r.m.s deviation root mean square)

-ITC table: please provide the error estimates from the fit and also the stoichiometry.

-Hydrogen bonds: please cite the software used for the analysis.

Reviewer #2 (Remarks to the Author):

This manuscript describes regulatory mechanisms by which regulate antagonistic Mei2 and Mmi1 in accumulation or degradation of meiotic RNAs during the mitosis-to-meiosis switching. The authors identified a new long noncoding RNA, mamRNA, that binds to Mei2 and Mmi1 and tunes their mutual control. This scheme of fine tuning of antagonistic proteins by lncRNA will provide a new insight into regulatory mechanisms for switching cell fates. Experiments were carefully executed by combining genetic/genomic, structural and biochemical approaches. Conclusions are supported by data presented. I have no serious concerns, but only have a few comments to improve the manuscript.

Major comments

1. Lines 94-199: First paragraph of Results may need more explanations about experimental designs to understand the conclusion. It is not obvious for readers how amounts of ssm4 and mcp5 are related to Mei2 down-regulation without knowing that these transcripts are regulated by Mmi1.

2. Lines 101-106: This paragraph again needs more explanations about experimental designs. The authors use mot2Δ mei2Δ double deletion cells to assess functionality of the Mei2 fragments, but expected phenotypes of mot2Δ cells in the absence or presence of Mei2 are not obvious for readers.

3. Discussion often repeats Results. Some of the discussion directly linked to specific results can be described in the Results sections, keeping more fundamental issues in Discussion.

Minor comments

4. Lines 617-618: It would be better to rephrase 5E-2, 1E-2, and 1E-3 by 0.05, 0.01, and 0.001, respectively.

Reviewer #3 (Remarks to the Author):

Review of "A scaffold lncRNA shapes the mitosis to meiosis switch" by Andric et al.

Multiple layers of regulation ensure that the switch from mitotic cell proliferation to the meiotic program is tightly controlled. Entry into meiosis requires the upregulation of a plethora of genes whose expression is normally repressed by multiple machineries including Mmi1, a YTH domain containing protein that associates with the RNA degradation activity of the exosome and RNAi machinery to degrade DSR containing transcripts and assemble repressive heterochromatin. Mmi1 itself is involved in a reciprocal regulatory circuit with the Mei2 protein and meiRNA, which are kept at low levels in mitotic cells in an Mmi1 dependent manner but are upregulated and sequester Mmi1 during meiosis. Exactly how the mutual dependence of Mmi1 and Mei2 is established and controlled has remained unclear. Here the authors report that a regulatory lncRNA called mamRNA is a critical component of the molecular mechanism that allows reciprocal regulation of Mmi1 and Mei2 during mitosis.

Overall this is an impressive study. The authors have used an excellent combination of genetics, structural biology and cell biology to show that a long non-coding RNA named mamRNA serves as a scaffold to promote mutual control by the antagonistic RNA binding proteins Mmi1 and Mei2. The conclusions presented are interesting. However, this reviewer has questions about the experimental conditions, which as described below have important implications for the interpretation of results.

Specific comments:

The authors need to provide details regarding the conditions used to perform the experiments. Growth of homothallic cells in minimal medium (EMM), which has been described in the methods section, usually leads to cell mating and meiosis. Considering that Mei2 distributes differently during mitosis (in cytoplasm) and meiosis (in nucleus) (Yamashita et al., 1998, Cell), it seems important to know the cell state (mitotic, meiotic or mixture). Knowing this answer is critical for interpreting the results since Mei2 and mamRNA are expected to localize to two different cellular compartments (mamRNA in nucleus and Mei2 in cytoplasm) in mitotic cells.

Supplementary Fig. 1c,d: the authors argue that Mei2-RRM3 Δ fails to accumulate meiotic mRNAs, suggesting that Mei2 RNA-binding activity is required to inhibit Mmi1 function. Conceptually, I agree with this argument. However, an important consideration is that if Mei2-RRM3 Δ is mainly localized to the cytoplasm, then of course it cannot inhibit Mmi1 function.

Fig. 1g: the authors shall show the cellular localization of Mei2-F644A. If the mutant protein is not in the nucleus, it obviously cannot be targeted by the ubiquitination machinery linked to mamRNA-Mmi1 in the nucleus. A similar concern applies to the results of the RNA-IP experiments.

Fig. 2a: the seqRIP-seq result suggests that meiotic cells are used because the authors identified meiRNA, which is upregulated specifically during meiosis; and also, Mei2 would likely have entered the nucleus to associate with Mmi1-bound RNA. If this is true, then the authors identified the mamRNA in meiotic cells. It is reasonable to believe that mamRNA works redundantly for Mmi1 sequestration with meiRNA. As shown in Fig. 3e, without meiRNA, the mamRNA is still able to block Mmi1 activity to promote meiosis, although with much less efficiency (only 8.6% with Mei2 overproduction), suggesting mamRNA plays only a minor role as compared with meiRNA. On the other hand, it is hard to believe that mamRNA plays a major role in blocking Mei2 activity in mitotic cells. Considering that Mei2 is controlled by Pat1 and Tor2 and is mainly distributed in the cytoplasm during mitosis, it is unclear how nuclear localized mamRNA would block Mei2 function. Do Mmi1, Mei2 and mamRNA colocalize in mitotic cells, especially in the mot2 Δ cells? The author shall consider showing Mei2 distribution in Supplementary Fig. 4a. This seems essential to support the major argument of this paper.

Interestingly, in Fig. 3e, the authors used Mei2-NLS to induce meiosis, suggesting that they are aware of the differences in Mei2 localization during mitosis and meiosis. The possibility that the differences between WT Mei2 and Mei2 mutants may not simply be due to defective RNA binding capacity, but

rather to the different cellular distributions, needs to be ruled out.

As mamRNA promotes Mei2 degradation through Ub-E3 ligase Mot2 (Fig. 3a,b), evidence of decreased Mei2 ubiquitination in mamRNA Δ cells can be included.

Can Mei2-RRM3 Δ or Mei2-F644A block meiosis, and if so, can the defect be rescued by mmi1 mutants? This would be interesting to know and is essential for supporting the model in this paper.

Minor points:

Mei2 and Mmi1 proteins are expressed under the control of the Pnmt41 inducible promoter, however it is unclear if the observed differences are due to differences in induction in different strains. For some key experiments, such as Fig.1d, g and Fig. 3a, b, RT-qPCR can be used to show that mei2 or mmi1 are expressed at comparable levels.

Can mamRNA overproduction rescue the meiotic defect of sme2 Δ , similar to the overexpression of the DSR regions of regulon genes (Harigaya et al., 2006, Nature)?

Response to reviewers' comments:

We are grateful to the reviewers for their insightful comments. We have revised the manuscript and performed additional experiments/controls, which further support the main conclusions of our work. Our point-by-point responses to the reviewers' remarks are shown below in blue text.

Reviewer #1 (Remarks to the Author):

This is a very interesting paper in which the authors identify a lncRNA that is important for switching from mitosis to meiosis. The authors identify the mamRNA that binds two crucial factors, Mmi1 and Mei2, which form a regulatory network.

Mmi1 is an RNA-binding protein associated with two major RNA degradation complexes, MTREC/Exosome, and the CCR4-NOT deadenylation complex. Mei2 was previously shown to bind another ncRNA, meiRNA, and this meiRNA-Mei2 RNP acts as a sponge to sequester Mmi1. The meiRNA, Mmi1 and Mei2 together form a regulatory circuit that controls the Mmi1 and Mei2 levels and thereby regulates RNA degradation. With the newly identified mamRNA, the authors add another regulator to this circuit.

The experiments are overall of good quality and also the figures are quite clear.

In general, it would be helpful for the reader if the experiments are introduced a little bit more detailed. For example, it is not clear why the authors use a mei4 deletion strain throughout their work. Please explain this. Similarly, the Mmi1 mutants defective for RNA-binding are introduced without explanation of how the residues for mutation were chosen (also see below).

We thank the reviewer for the positive comments. As requested, we have further introduced the experimental strategies and explained specific points to ease reading.

- Deletion of *mmi1*⁺ results in severe growth defects due to the ectopic expression of the meiosis-specific transcription factor Mei4 and its target genes (PMID 16823445). To suppress viability defects and avoid indirect effects, the analysis of mutants defective for Mmi1 function is generally carried out in cells deleted for Mei4. This information was solely mentioned in the Methods section and we have now included it early in the Results.

- Mmi1 mutants defective for RNA-binding (i.e. Mmi1-Y352F and Mmi1-Y466F) were selected based on previously published ITC experiments showing that the association of recombinant YTH^{Mmi1-Y352F} and YTH^{Mmi1-Y466F} domains to UUAAC-containing synthetic RNAs is severely affected (PMID 26673708). We have now added this information in the Results section.

More specific:

The center of this paper is the identification of the mamRNA, which contains predicted binding sites for Mmi1 and Mei2.

As this RNA is the most critical molecule of this paper, it deserves more attention. Conceptually, the mamRNA and the meiRNA are similar because both can bind Mmi1 and Mei2, but the mamRNA can obviously do something the meiRNA cannot. The mechanistic question is: Which features of the mamRNA make it a stronger regulator than the meiRNA.

As the paper is in several parts mechanistic/atomistic, I think it would be beneficial to

-Show the mamRNA sequence schematically (secondary structure/folding prediction) and highlight the positions of Mei2 (UU(U/G)G) and Mmi1 (UNAAAC) binding motifs in this scheme.

How does the mamRNA compare to the meiRNA, both predicted folding and the presence/number of Mei2 and Mmi1 binding motifs?

Would this maybe help to explain why one RNA escapes degradation during mitosis in contrast to meiRNA? Is it merely more structured or stable?

- We have included linear schemes of the mamRNA, meiRNA and omt3 lncRNAs (new Fig. 2c) with the positions of the Mmi1 and putative Mei2 binding motifs (i.e. UNAAAC and UU(U/G/A)G, respectively). Relative to meiRNA, mamRNA contains much less Mmi1 binding sites (3 versus 25) while Mei2 motifs display a similar density (16 sites in 678 nts versus 34 sites in 1562 nts) and are evenly distributed along transcripts. It should be noted that Mei2 only associates with the 5' region of meiRNA (PMID 24920274), implying that additional elements contribute to its binding to RNA. Whether this relates to longer binding sites and/or structured regions remains to be investigated, as now discussed in the Results section.
- We have also used structure prediction programs (e.g. mFOLD), which revealed a large number of possible folds with similar Gibbs free energy. Since this makes it difficult to select arbitrarily one over the others, and may not be informative as to how lncRNAs fold *in vivo*, we have decided not to include these data in the manuscript. In the future, dedicated experiments might shed light on the structures adopted by mamRNA and meiRNA *in cellulo*.
- The reason why mamRNA is not targeted for Mmi1-dependent degradation as opposed to meiRNA and other meiotic mRNAs remains unclear at this stage. The low number of UNAAAC motifs may explain why mamRNA escapes degradation, although previous work suggested that a single Mmi1 binding site is sufficient to promote RNA decay (PMID 26670050). Therefore, as suggested by the reviewer and mentioned in the discussion, the existence of structured regions and/or protein partners precluding exonucleolytic attack by the exosome, akin to sn/snoRNAs, may provide a rationale for mamRNA stability during mitosis.

The authors show at the very beginning that the mutations/deletions of RNA-binding domains of Mmi1 and Mei2 abrogate the mutual Mmi1/Meo2 regulatory system. Later they show that the deletion of the mamRNA phenocopies the mot2 or mei4/mmi1 deletion strains for Mei2-levels. As the binding motif for the Mmi1 YTH motif is known, and the authors here identify the Mei2 binding motif, it would have been fantastic (and even more mechanistic) if the authors would have deleted or mutated these motifs within the mamRNA (or perform rescue experiments with different mamRNA constructs).

We thank the reviewer for pointing out these excellent suggestions. We agree that identifying the *cis*-elements within mamRNA that are involved in the binding and regulations of both Mmi1 and Mei2 would provide additional mechanistic insights. However, not all UNAAAC motifs are bound by Mmi1, suggesting that specific RNA folds and/or protein partners may contribute to binding (PMID 26670050). Furthermore, putative Mei2 motifs (i.e. UU(U/G/A)G) spread all along mamRNA and meiRNA (new Fig. 2c), while Mei2 only associates with the 5' region of the latter (PMID 24920274; see above). Other sequence and/or structure elements are thus very likely to determine binding specificity and their identification would require extensive additional work, which we believe is outside the scope of the present study.

How did the authors find the similarity to Sxl RRM1? Does this come from a PDB wide comparison e.g., Dali search?

The structural similarity was indeed found from a PDB wide comparison using Dali server and best hits were screened for those containing an RNP1 motif with the GYAF sequence similarly to Mei2. This is now explained in the revised manuscript (Results section).

Please show the ITC titrations. A table with the thermodynamic data does not allow to judge the quality of the experiment. It is also not clear how the data were fitted, as the stoichiometry n is not indicated. Was n fixed to 1, after determination of the 1:1 stoichiometry from MALLS? Please also list the error limits from the estimated fit.

The ITC titration curves have been added to the revised manuscript (Fig. 1e, Supplementary Fig. 1g) and the table describing the thermodynamics values from the ITC experiments have been updated as requested (Supplementary Table 2). Initially, the data were fitted with a one binding site model and stoichiometries were not fixed to 1 for the fitting. The stoichiometries determined from the fitting were around 0.5, most probably because the RNA used in these experiments was only partially functional and/or degraded. Following the reviewer's comment and considering the 1:1 stoichiometry determined by SEC-MALLS experiments, we have reanalyzed the ITC data and processed them by fixing stoichiometries to 1. As a result, the thermodynamic parameters have slightly changed but this does not impact our initial conclusions, i.e. RNAs with one or two central Gs interact more tightly with RRM3^{Mei2} than a poly-U₁₅.

The structural data look very solid and, together with the ITC data, allow to suggest an RNA-binding motif recognized by the Mei2 RRM3. However, I am not entirely convinced about the UU(U/G)G motif. Changing one internal U to a G increased the affinity by a factor of 7. Replacing an upstream U by a G then again reduces the affinity by a factor of 2. This clearly shows how the preference of UUUG > UUGG > UUUU. The nucleotide at position seven does not form many hydrogens bonds, so how do the authors exclude the possibility of the presence of A or C at position 7? Could the motif also be UUNG?

The U at position 7 forms a single hydrogen bond between its O4 atom and the NH group of N680. Such hydrogen bond would not be possible with a C, given the presence of an amino group at the corresponding position of the ring. This is the reason why we ruled out a C at position 7 of the motif. Regarding A, we agree that there is no structural reason to exclude it from the motif, which then should be UU(U/G/A)G. This is now explained in more details in the revised version of the manuscript (Results section).

Line 115: 'recognition of a G11 by Slx1 F170' – I agree that the F can participate pi-stacking with the U base, but base-specific recognition is usually realized by reading out hydrogen bonding patterns (e.g., see S1g). Please change 'recognize' to 'interact'.

This is indeed correct. We have modified as requested.

Minor points

Supplementary Figure 1b - Hoescht should read Hoechst

We have corrected this.

Supplementary Figure 1f – the unit for the refractive index is not mAU (Milli absorption units). Do the authors mean arbitrary units? Please specify. It would help note the mass of the U15 RNA so that the theoretical mass of the complex can be compared to the measured.

We thank the reviewer for noticing this mistake. We indeed meant Arbitrary Units. We have corrected this in the revised manuscript and also included the information regarding the molecular weight of the RNA (4.29 kDa) in the figure legend.

Line 113 - MALLS not explained, but SEC is.

We have corrected this.

Supplementary Table 1: please divide the one table as these are three different things and add the appropriate descriptions required to understand the table.

As requested, we have divided the relevant information in 3 different tables.

- Data collection and refinement (r.m.s deviation root mean square)
- ITC table: please provide the error estimates from the fit and also the stoichiometry.

This information has been added.

- Hydrogen bonds: please cite the software used for the analysis.

The hydrogen bonds were detected using the PISA server and validated by careful analysis of the electron density maps. This has been specified in the revised version of the manuscript (Results section).

Reviewer #2 (Remarks to the Author):

This manuscript describes regulatory mechanisms by which regulate antagonistic Mei2 and Mmi1 in accumulation or degradation of meiotic RNAs during the mitosis-to-meiosis switching. The authors identified a new long noncoding RNA, mamRNA, that binds to Mei2 and Mmi1 and tunes their mutual control. This scheme of fine tuning of antagonistic proteins by lncRNA will provide a new insight into regulatory mechanisms for switching cell fates. Experiments were carefully executed by combining genetic/genomic, structural and biochemical approaches. Conclusions are supported by data presented. I have no serious concerns, but only have a few comments to improve the manuscript.

We thank the reviewer for the overall positive comments. We have modified the text as requested to improve the manuscript.

Major comments

1. Lines 94-199: First paragraph of Results may need more explanations about experimental designs to understand the conclusion. It is not obvious for readers how amounts of *ssm4* and *mcp5* are related to Mei2 down-regulation without knowing that these transcripts are regulated by Mmi1.

We have reformulated the text to make clear that the *ssm4+* and *mcp5+* DSR-containing meiotic mRNAs are targeted by Mmi1. As such, their abundance increases upon accumulation of Mei2 in *mot2Δ* cells, due to Mmi1 inactivation.

2. Lines 101-106: This paragraph again needs more explanations about experimental designs. The authors use *mot2Δ mei2Δ* double deletion cells to assess functionality of the Mei2 fragments, but expected phenotypes of *mot2Δ* cells in the absence or presence of Mei2 are not obvious for readers.

We have rephrased the text accordingly. The absence of Mot2 leads to increased Mei2 levels, which in turn impact Mmi1 activity and trigger meiotic mRNA accumulation. Hence, we used *mot2Δ mei2Δ* cells, in which meiotic mRNA degradation by Mmi1 is restored due to the absence of its inhibitor Mei2, to assess the impact of plasmid-borne Mei2 variants on the abundance of meiotic transcripts.

3. Discussion often repeats Results. Some of the discussion directly linked to specific results can be described in the Results sections, keeping more fundamental issues in Discussion.

We have followed the reviewer's advice and emphasized conceptual issues in the Discussion.

Minor comments

4. Lines 617-618: It would be better to rephrase 5E-2, 1E-2, and 1E-3 by 0.05, 0.01, and 0.001, respectively.

As suggested, we have rephrased this.

Reviewer #3 (Remarks to the Author):

Review of “A scaffold lncRNA shapes the mitosis to meiosis switch” by Andric et al.

Multiple layers of regulation ensure that the switch from mitotic cell proliferation to the meiotic program is tightly controlled. Entry into meiosis requires the upregulation of a plethora of genes whose expression is normally repressed by multiple machineries including Mmi1, a YTH domain containing protein that associates with the RNA degradation activity of the exosome and RNAi machinery to degrade DSR containing transcripts and assemble repressive heterochromatin. Mmi1 itself is involved in a reciprocal regulatory circuit with the Mei2 protein and meiRNA, which are kept at low levels in mitotic cells in an Mmi1 dependent manner but are upregulated and sequester Mmi1 during meiosis. Exactly how the mutual dependence of Mmi1 and Mei2 is established and controlled has remained unclear. Here the authors report that a regulatory lncRNA called mamRNA is a critical component of the molecular mechanism that allows reciprocal regulation of Mmi1 and Mei2 during mitosis.

Overall this is an impressive study. The authors have used an excellent combination of genetics, structural biology and cell biology to show that a long non-coding RNA named mamRNA serves as a scaffold to promote mutual control by the antagonistic RNA binding proteins Mmi1 and Mei2. The conclusions presented are interesting. However, this reviewer has questions about the experimental conditions, which as described below have important implications for the interpretation of results.

We are grateful to the reviewer for the positive feedback. We have addressed the concerns as discussed below.

Specific comments:

The authors need to provide details regarding the conditions used to perform the experiments. Growth of homothallic cells in minimal medium (EMM), which has been described in the methods section, usually leads to cell mating and meiosis. Considering that Mei2 distributes differently during mitosis (in cytoplasm) and meiosis (in nucleus) (Yamashita et al., 1998, Cell), it seems important to know the cell state (mitotic, meiotic or mixture). Knowing this answer is critical for interpreting the results since Mei2 and mamRNA are expected to localize to two different cellular compartments (mamRNA in nucleus and Mei2 in cytoplasm) in mitotic cells.

- We indeed used homothallic strains grown in minimal medium, which eventually commit to meiosis when starved for nutrients (i.e. when reaching high optical density, $OD > 1.5-2.0$). However, in all our experiments, cells were maintained in exponential phase of growth by successive culture dilutions prior to harvesting (i.e. $0.4 < OD < 1.0$). Hence, unless otherwise specified (smFISH analyses in Supplementary Fig. 2h and sporulation assays in Fig. 3e and Supplementary Fig. 3e,f), we systematically assessed the role of mamRNA in mitotic cells.

- About the cellular localization of Mei2, we previously showed that the endogenous GFP-tagged protein is barely detectable, if at all, in wild type mitotic cells due to extensive degradation (PMID 28841135). However, in the absence of the E3 ubiquitin ligases Mot2 or Ubr1 that both control its abundance, Mei2 accumulates throughout the cell, both in the nucleus and the cytoplasm (PMID 28841135). In addition, it was earlier demonstrated that Mei2 shuttles between the nucleus and the cytoplasm in mitotic cells (PMID 11423126; see below). Thus, our results that mamRNA and Mmi1 target a pool of Mei2 in the nucleus are compatible with their respective cellular distribution. This has now been clarified in the Results section.

Supplementary Fig. 1c,d: the authors argue that Mei2-RRM3Δ fails to accumulate meiotic mRNAs, suggesting that Mei2 RNA-binding activity is required to inhibit Mmi1 function. Conceptually, I

agree with this argument. However, an important consideration is that if Mei2-RRM3 Δ is mainly localized to the cytoplasm, then of course it cannot inhibit Mmi1 function.

Please see our answer to the next point.

Fig. 1g: the authors shall show the cellular localization of Mei2-F644A. If the mutant protein is not in the nucleus, it obviously cannot be targeted by the ubiquitination machinery linked to mamRNA-Mmi1 in the nucleus. A similar concern applies to the results of the RNA-IP experiments.

We have now included microscopy data on the cellular localization of the Mei2-F644A mutant. As shown in new Supplementary Fig. 1m, plasmid-borne wild type and mutant Mei2 distribute throughout the cell, localizing both in the cytoplasm and the nucleus. This is consistent with our previous work and data showing that the two versions of the protein shuttle between the nucleus and the cytoplasm (PMID 28841135; PMID 11423126). We have mentioned such information in the Introduction and Results sections to remove any ambiguity. Thus, defective Mei2-F644A downregulation and binding to lncRNAs, including mamRNA, does not merely reflect impaired nuclear localization.

Fig. 2a: the seqRIP-seq result suggests that meiotic cells are used because the authors identified meiRNA, which is upregulated specifically during meiosis; and also, Mei2 would likely have entered the nucleus to associate with Mmi1-bound RNA. If this is true, then the authors identified the mamRNA in meiotic cells. It is reasonable to believe that mamRNA works redundantly for Mmi1 sequestration with meiRNA. As shown in Fig. 3e, without meiRNA, the mamRNA is still able to block Mmi1 activity to promote meiosis, although with much less efficiency (only 8.6% with Mei2 overproduction), suggesting mamRNA plays only a minor role as compared with meiRNA. On the other hand, it is hard to believe that mamRNA plays a major role in blocking Mei2 activity in mitotic cells. Considering that Mei2 is controlled by Pat1 and Tor2 and is mainly distributed in the cytoplasm during mitosis, it is unclear how nuclear localized mamRNA would block Mei2 function. Do Mmi1, Mei2 and mamRNA colocalize in mitotic cells, especially in the *mot2* Δ cells? The author shall consider showing Mei2 distribution in Supplementary Fig. 4a. This seems essential to support the major argument of this paper.

- The seqRIP-seq approach was carried out from mitotic cells over-expressing tagged Mei2, and thereby accumulating DSR-containing meiotic RNAs such as meiRNA, due to Mmi1 inactivation (PMID 28841135). This is most likely the reason why meiRNA was found among the most enriched and abundant RNA species simultaneously bound by Mmi1 and Mei2. The association of Mei2 to Mmi1-bound transcripts in this context also supports the notion that the former localizes to the nucleus in mitotic cells. Further, if Mei2 was strictly cytoplasmic, then its increased levels in the absence of Mot2 may not inhibit Mmi1 nor lead to the accumulation of meiotic RNAs, which is not what we observed (PMID 28841135; this study). These experimental details regarding the seqRIP-seq procedure were originally described in the figure legend and Methods but we have now introduced them in the Results section to avoid any confusion.

- Our data support a model whereby Mmi1 associates with mamRNA to target a pool of nuclear Mei2 to Mot2 in mitotic cells. The function of mamRNA is therefore essential to maintain low levels of Mei2 during mitosis and hence preserve the function of Mmi1 in meiotic mRNA degradation. Conversely, when Mei2 downregulation is impaired in *mot2* Δ mitotic cells, mamRNA is necessary for the inhibition of Mmi1 by increased Mei2 levels. We previously showed that Mei2 accumulates throughout the cell in the absence of Mot2 (PMID 28841135), implying that a fraction of the protein colocalizes with Mmi1 and mamRNA in this context. Our results also indicate that, contrary to meiRNA, mamRNA is not required for meiosis progression *per se* (Supplementary Fig. 3e). Nonetheless, mamRNA can take over meiRNA at low frequency if Mei2 is artificially tethered to the nucleus (Fig. 3e), suggesting that the role of mamRNA in the inhibition of Mmi1 may contribute to

the initiation of sexual differentiation. Thus, *mamRNA* has a major role in the reciprocal control of Mmi1 and Mei2 activities in the nucleus of mitotic cells and may facilitate Mmi1 inactivation upon meiosis onset. This has been rephrased accordingly in the Discussion section.

Interestingly, in Fig. 3e, the authors used Mei2-NLS to induce meiosis, suggesting that they are aware of the differences in Mei2 localization during mitosis and meiosis. The possibility that the differences between WT Mei2 and Mei2 mutants may not simply be due to defective RNA binding capacity, but rather to the different cellular distributions, needs to be ruled out.

Previous work showed that Mei2-NLS suppresses the sporulation defects observed in the absence of *meiRNA* (i.e. *sme2Δ* cells) (PMID 9778252). Crucially, however, expression of Mei2-F644A-NLS failed to do so, indicating that Mei2 RNA-binding capacity is an absolute requirement even when the protein is targeted to the nucleus (PMID 9778252). Thus, and in agreement with our microscopy data (new Supplementary Fig. 1m) and the literature (PMID 11423126; see above), defective regulations in the Mei2-F644A mutant do not simply reflect different cellular distributions but rather highlight the importance of Mei2 RNA-binding activity.

As *mamRNA* promotes Mei2 degradation through Ub-E3 ligase Mot2 (Fig. 3a,b), evidence of decreased Mei2 ubiquitination in *mamRNAΔ* cells can be included.

We previously showed that Mot2 targets a pool of Mei2 for ubiquitinylation to limit its accumulation in mitotic cells (PMID 28841135). In this study, we demonstrate that Mmi1 associates with *mamRNA* to target Mei2 for downregulation by Mot2, which is essential to safeguard Mmi1 activity in meiotic mRNA degradation during mitotic growth. Importantly, we now provide evidence that *mot2Δ* and *mamRNAΔ* are epistatic with respect to Mei2 protein levels, since the absence of one or both genes leads to a similar increase in Mei2 levels (new Fig. 3a). In addition, *mamRNA* is specifically required for the accumulation of Mmi1 RNA targets upon deletion of *mot2+* (Fig. 3c,d, Supplementary Fig. 3d), as other upregulated genes that do not belong to the Mmi1 regulon still exhibit increased expression in *mot2Δ mamRNAΔ* cells (Supplementary Fig. 3d). Together, these findings strongly support the notion that Mmi1, Mot2 and *mamRNA* function in the same pathway to lower Mei2 abundance. As per this model, the absence of *mamRNA* is expected to result in decreased Mei2 ubiquitinylation, akin to *mot2Δ* cells.

Can Mei2-RRM3Δ or Mei2-F644A block meiosis, and if so, can the defect be rescued by *mmi1* mutants? This would be interesting to know and is essential for supporting the model in this paper.

Previous work showed that Mei2-F644A blocks meiosis at an early step, prior to pre-meiotic DNA synthesis (PMID 7520368). However, we could not test whether this defect can be suppressed by Mmi1 YTH mutants as the latter were generated from a parental strain lacking Mei4, a meiosis-specific transcription factor essential for sexual differentiation and sporulation. This is because Mmi1 mutants themselves exhibit severe viability defects due to the ectopic expression of Mei4 (*mei4+* mRNAs are normally degraded in an Mmi1-dependent manner during mitotic growth) and its target genes (see also our answer to reviewer #1). Nonetheless, given that Mmi1 YTH mutants fail to associate with *meiRNA* (Fig. 2f), which is instead mandatory for meiosis progression (PMID 24920274), we believe it is unlikely they would rescue Mei2-F644A meiotic defects and *vice versa*.

Minor points:

Mei2 and Mmi1 proteins are expressed under the control of the Pnmt41 inducible promoter, however it is unclear if the observed differences are due to differences in induction in different strains. For some key experiments, such as Fig. 1d, g and Fig. 3a, b, RT-qPCR can be used to show that *mei2* or *mmi1* are expressed at comparable levels.

- As requested, we performed RT-qPCR assays to determine *mmi1*⁺ and *mei2*⁺ mRNA levels produced from the *nmt41* promoter. *mei4*Δ *mmi1*Δ P_{nmt41}-TAP-Mei2 cells expressing plasmid-borne wild type Mmi1 or YTH mutants exhibited very similar *mei2*⁺ mRNA levels (maximum 1.6-fold increase; new Supplementary Fig. 1a; related to Fig. 1d), strongly suggesting that higher amounts of Mei2 protein in Mmi1 mutants do not simply result from an increased expression of the *mei2*⁺ gene. We further detected higher levels of *mmi1*-Y352F and *mmi1*-Y466F transcripts when compared to *mmi1*⁺ mRNAs (new Supplementary Fig. 1a), which correlated with an increased abundance of the mutant proteins (see panel α-FLAG in Fig. 1d). Yet, this was not sufficient to lower Mei2 protein levels as wild type Mmi1 does, highlighting the importance to preserve the integrity of the YTH domain. Whether increased mRNA and protein levels of Mmi1 YTH mutants underlie a role for the protein in targeting its own transcript for degradation or other indirect effects remains to be elucidated.

- We also observed that plasmid-borne *mei2*-F644A mRNA levels were about 3-fold higher than wild type *mei2*⁺ mRNAs (new Supplementary Fig. 1l; related to former Fig. 1g). Nonetheless, we estimated that the mutant protein was roughly 10-fold more abundant than its wild type counterpart (new Fig. 1i). Thus, while some of the Mei2-F644A protein accumulation may result from increased expression, defective RNA-binding capacity most likely leads, for a large share, to higher protein levels. To remove any ambiguity, we added to the revised manuscript these new data (new Fig. 1i and Supplementary Fig. 1l), which still support our model that Mei2 binding to mamRNA is required for its downregulation by Mmi1 and Mot2 in mitotic cells. Consistent with this notion, deletion of Mmi1, Mot2 or mamRNA strongly increases Mei2 protein levels while marginally affecting, if at all, *mei2*⁺ mRNA levels produced from the *nmt41* promoter (maximum 1.7-fold increase in *mot2*Δ cells) (new Supplementary Fig. 3b; related to Fig. 3a,b).

Can mamRNA overproduction rescue the meiotic defect of *sme2*Δ, similar to the overexpression of the DSR regions of regulon genes (Harigaya et al., 2006, Nature)?

We have assessed whether overexpression of mamRNA can suppress sporulation defects in *sme2*Δ cells. However, we did not observe a substantial increase in spore formation (new Supplementary Fig. 3f), indicating that higher mamRNA levels are not sufficient to lure Mmi1 upon meiosis onset, as DSR elements do (PMID 16823445; PMID 24920274). This might relate to the lower number of UNAAAC motifs within mamRNA compared to *bona fide* meiotic mRNA targets (see new Fig. 2c).

REVIEWERS' COMMENTS

Reviewer #1 (Remarks to the Author):

The authors have addressed all my points and I would suggest that the manuscript can be accepted/published in this revised form.

The only minor comment:

The new Figure 2C showing the linear models of the RNA is very helpful, but the colors with red and green are not a good combination. Please change this so people with impaired colour vision also can discriminate the two motifs.

Reviewer #3 (Remarks to the Author):

I have read the revised version of the manuscript by Andric et al. In general, the authors have done good job responding to this reviewers' comments. However, I have few queries related to the author's responses.

In response to a comment about the seqRIP-seq results, the authors commented, "The seqRIP-seq approach was carried out from mitotic cells over-expressing tagged Mei2, and thereby accumulating DSR-containing meiotic RNAs such as meiRNA, due to Mmi1 inactivation (PMID 28841135)." The details of the Mei2 expression shall be provided. This is important because in a previous paper by the authors, mei2 expression from thiamine repressible nmt41 promoter was not sufficient to induce meiotic gene expression (PMID 28841135).

They further commented, "...if Mei2 was strictly cytoplasmic, then its increased levels in the absence of Mot2 may not inhibit Mmi1 nor lead to the accumulation of meiotic RNAs, which is not what we observed (PMID 28841135; this study)." I wonder whether the authors have ruled out the possibility that loss of Mot2 causes defects in CCR4-NOT mediated elimination of meiotic RNAs. Do they know if the loss of other subunits of the CCR4-NOT deadenylation complex (which are not required for Mei2 degradation (PMID 28841135) also affects the accumulation of meiotic RNAs.

"...We previously showed that Mei2 accumulates throughout the cell in the absence of Mot2 (PMID 28841135), implying that a fraction of the protein colocalizes with Mmi1 and mamRNA in this context." If mamRNA is not able to sequester Mei2, then how does it controls the Mei2 levels? Comment.

"we could not test whether this defect can be suppressed by Mmi1 YTH mutants as the latter were generated from a parental strain lacking Mei4, a meiosis specific transcription factor essential for sexual differentiation and sporulation. This is because Mmi1 mutants themselves exhibit severe viability defects due to the ectopic expression of Mei4 (mei4+ mRNAs are normally degraded in an Mmi1-dependent manner during mitotic growth) and its target genes". The authors' could have used mmi1-619 mutant allele that can be grown without deleting mei4.

RESPONSES TO REVIEWERS' COMMENTS

We thank the reviewers for their comments and suggestions. Our point-by-point responses are found below in blue text.

Reviewer #1 (Remarks to the Author):

The authors have addressed all my points and I would suggest that the manuscript can be accepted/published in this revised form.

The only minor comment:

The new Figure 2C showing the linear models of the RNA is very helpful, but the colors with red and green are not a good combination. Please change this so people with impaired colour vision also can discriminate the two motifs.

We are grateful to the reviewer for the positive comments. As requested, we have modified the colors of Mmi1 and Mei2 binding sites in Figure 2c.

Reviewer #3 (Remarks to the Author):

I have read the revised version of the manuscript by Andric et al. In general, the authors have done good job responding to this reviewers' comments. However, I have few queries related to the author's responses.

We thank the reviewer for the positive comments.

In response to a comment about the seqRIP-seq results, the authors commented, "The seqRIP-seq approach was carried out from mitotic cells over-expressing tagged Mei2, and thereby accumulating DSR-containing meiotic RNAs such as meiRNA, due to Mmi1 inactivation (PMID 28841135)." The details of the Mei2 expression shall be provided. This is important because in a previous paper by the authors, mei2 expression from thiamine repressible nmt41 promoter was not sufficient to induce meiotic gene expression (PMID 28841135).

We thank the reviewer for pointing this out. Three versions of the nmt promoter, namely nmt81, nmt41 and nmt1 are associated with low, mild and strong expression, respectively. We indeed previously showed that *mei2*⁺ expression from the nmt41 promoter does not lead to the accumulation of meiotic mRNAs (Figure 3D in PMID 28841135). However, the seqRIP-seq approach in the present study was performed using cells over-expressing *mei2*⁺ with the nmt1 promoter, which results in a pronounced increase in the levels of Mmi1 RNA targets (Figure 4B in PMID 28841135). This information was only mentioned in the legend of supplementary figure 2b and we have now included it in the legend of figure 2a.

They further commented, "...if Mei2 was strictly cytoplasmic, then its increased levels in the absence of Mot2 may not inhibit Mmi1 nor lead to the accumulation of meiotic RNAs, which is not what we observed (PMID 28841135; this study)." I wonder whether the authors have ruled out the possibility that loss of Mot2 causes defects in CCR4-NOT mediated elimination of meiotic RNAs. Do they know if the loss of other subunits of the CCR4-NOT deadenylation complex (which are not required for Mei2 degradation (PMID 28841135)) also affects the accumulation of meiotic RNAs.

In our previous study (PMID 28841135), we showed that among non-essential subunits of the Ccr4-Not complex (that is all but the scaffolding subunit Not1), only Mot2 is required to restrict the levels of Mei2 and hence prevent the accumulation of Mmi1 RNA targets. Individual deletions of the RNA deadenylases Ccr4 and Caf1 or of the Not2, Not3 and Rcd1 subunits do not impact the levels of meiotic mRNAs (nor Mei2 abundance) (Figure 2B in PMID 28841135). We have included this information in the Introduction.

"..We previously showed that Mei2 accumulates throughout the cell in the absence of Mot2 (PMID 28841135), implying that a fraction of the protein colocalizes with Mmi1 and mamRNA in this context." If mamRNA is not able to sequester Mei2, then how does it controls the Mei2 levels? Comment.

Our results support a model whereby Mmi1 associates with mamRNA to target Mei2 for ubiquitinylation and downregulation by Mot2. In other words, Mei2 interacts with mamRNA to restrict its own levels in wild type mitotic cells. Contrarily, the absence of Mot2 causes an accumulation of Mei2, which distributes throughout the cell. This implies that only a fraction of accumulating Mei2 remains bound to mamRNA to in turn inhibit Mmi1. It is possible that excess of Mei2 disperses in the nucleus and cytoplasm due to a limitation in available mamRNA binding sites and/or to a slow Mei2-mamRNA dissociation rate. We now mention these points in the Discussion.

"we could not test whether this defect can be suppressed by Mmi1 YTH mutants as the latter were generated from a parental strain lacking Mei4, a meiosis specific transcription factor essential for sexual differentiation and sporulation. This is because Mmi1 mutants themselves exhibit severe viability defects due to the ectopic expression of Mei4 (mei4+ mRNAs are normally degraded in an Mmi1-dependent manner during mitotic growth) and its target genes". The authors' could have used mmi1-619 mutant allele that can be grown without deleting mei4.

Unfortunately, we do not have the mmi1-619 mutant allele in our lab stock but we thank the reviewer for this suggestion. We will consider the use of this hypomorphic mutant in future analyses.